



# Radionuclide tracers reveal new Arctic pathways shaping water mass mixing and formation in Baffin Bay and Labrador Sea

Lisa G.T. Leist[1], Maxi Castrillejo[2], Kumiko Azetsu-Scott[3], Craig Lee[4], Jed Lenetsky[5], Marc Ringuette[3], Christof Vockenhuber[1,6], Habacuc Pérez-Tribouillier[1,6], and Núria Casacuberta[1,6]

[1]Institute of Biogeochemistry and Pollutant Dynamics, Department of Environmental Systems Science, ETH Zurich, Zurich, Switzerland
[2]Institute of Earth Sciences, University of Lausanne, Lausanne, Switzerland
[3]Bedford Institute of Oceanography, Dartmouth, Nova Scotia, Canada
[4]Applied Physics Laboratory, University of Washington, Seattle, Washington
[5]Department of Atmospheric and Oceanic Sciences and Institute of Arctic and Alpine Research, University of Colorado - Boulder, Boulder, Colorado, USA
[6]Laboratory of Ion Beam Physics, Department of Physics, ETH Zurich, Zurich, Switzerland

**Correspondence:** Lisa Leist (lisa.leist@usys.ethz.ch) and Núria Casacuberta (cnuria@ethz.ch)

**Abstract.** The Davis Strait is one of two key Arctic gateways, where waters derived from the Atlantic flow northward and exchange with Arctic-origin waters flowing southward. This interaction may play a crucial role in shaping the formation of deep water masses in the subpolar North Atlantic. This study employs observations from 2022 and 2024 of the two artificial radionuclides $^{129}$I and $^{236}$U measured in Baffin Bay, Davis Strait and the Labrador Sea. Samples were collected during three

5    expeditions: the AZOMP occupation of the AR7W Line in May 2022, the Davis Strait Observation Programme in October 2022, and the Amundsen Expedition as part of the Transforming Climate Action programme in September-October 2024. By defining the characteristic $^{129}$I and $^{236}$U concentrations of the main inflowing water masses (endmembers), we examined the distribution, origin and formation of key Baffin Bay water masses. This approach also allowed us to quantify the contribution of Transition Water to the formation of Labrador Sea Water (LSW) and North East Atlantic Deep Water (NEADW). Our results

10    reveal a substantial contribution of West Greenland Shelf Water to Arctic Water on the surface of central Baffin Bay, accounting for approximately 30%. High $^{236}$U identified a previously unknown pathway of Arctic-Atlantic-derived waters entering Baffin Bay via Lancaster Sound, contributing 40-50% to the formation of Transition Water. In contrast, cold Arctic Water appears to originate mainly from Nares Strait, with contributions of Arctic-Atlantic Water outflowing Nares Strait reaching up to 35%. Notably, the contribution of fresh Transition Water to the formation of LSW was significant, exceeding 30%. However, the

15    binary mixing model showed limitations in quantifying the origin of NEADW due to low tracer concentrations and the likely influence of multiple water mass sources. This study offers novel insights into the origin and transformation of waters in Baffin Bay and the Labrador Sea and enhances our understanding of the complex interactions between the Arctic Ocean and the subpolar North Atlantic.



## 1 Introduction

### 1.1 The Davis Strait, connecting the Arctic with the subpolar North Atlantic

Davis Strait, located between west Greenland and Baffin Island (Fig. 1A), plays a crucial role in the exchange of water masses between the Arctic and the North Atlantic oceans (Haine et al., 2015; Curry et al., 2014; Huang et al., 2024; Rudels et al., 2004; Komuro and Hasumi, 2005). The 1060 m deep sill of Davis Strait marks the southern boundary of Baffin Bay (max. depth of 2300 m), which receives approximately 2 Sv (1 Sv = $10^6$ m$^3$s$^{-1}$) of southward-flowing Arctic waters through multiple channels: from the central Arctic through Nares Strait, and from the Canada Basin through the Canadian Arctic Archipelago (CAA) (Münchow et al., 2015; Rudels, 2011; Tang et al., 2004; Pelle et al., 2024). Davis Strait also receives about 3 Sv of northward-flowing Atlantic water from the Labrador Sea. The Labrador Sea is an integral component of the subpolar gyre and a key deep convection area for the Atlantic Meridional Overturning Circulation (Tang et al., 2004; Le Bras, 2023; Lozier, 2023).

These evolving pathways and their associated variability underscore the importance of understanding the complex circulation system of the region. The general circulation, indicated in Fig. 1A, is largely influenced by boundary currents. East of Davis Strait, the northward flowing West Greenland Current (WGC, dark red arrows, Fig. 1A) transports West Greenland Shelf Water (WGSW) along the Greenland Shelf into Baffin Bay (Cuny et al., 2005; Curry et al., 2014; Huang et al., 2024). The WGC originates from the East Greenland Current (EGC, dark red arrows, Fig. 1A), which outflows Fram Strait southward along the Greenland coast, carrying fresh and relatively warm Polar Surface Water (PSW) from the Arctic Ocean (Sutherland et al., 2009). After rounding the southern tip of Greenland at Cape Farewell, the EGC becomes the WGC as it enters the Labrador Sea (Gou et al., 2022). In the northern Labrador Sea, it bifurcates into one branch continuing towards Baffin Bay and a second branch, following the bathymetry of the Labrador Sea, turning westwards to the Canadian Shelf (Myers et al., 2009). In the Labrador Sea, the WGC is joined by a branch of the North Atlantic Current (NAC, black arrow Fig. 1A), which transports saline West Greenland Irminger Water (WGIW, light green arrow Fig. 1A) originating from the subtropics (Cuny et al., 2002).

Western Davis Strait is dominated by the southward-flowing surface Baffin Island Current (BIC, orange arrow Fig. 1A), which transports fresh water of arctic origin, such as Arctic Water (AW) and cold Arctic Water (cold AW) along the Baffin Island margin to the Labrador Sea (Cuny et al., 2005). The Arctic Water is considered of Arctic origin with a strong influence of glacial runoff, air-sea fluxes, and sea ice melt (Curry et al., 2014), while the cold Arctic Water (similar to the hydrographic properties of cold Polar Water (Huang et al., 2024)) represents a subset of cold and more saline water within the Arctic Water, which may receive less freshwater and experience stronger cooling and salinification. After entering Baffin Bay via Nares Strait (220 m, Jackson et al. (2014); Rabe et al. (2010)), Lancaster Sound (125 m, Peterson et al. (2012)) and Jones Sound (125 m, Melling et al. (2008)), these water masses with low salinities (<33.5) and low temperatures (<1ºC) are present primarily at the surface of Baffin and along the BIC.

In central Davis Strait, at intermediate depths between 300 m and 600 m, relatively warm Transition Water (dark blue Fig. 1A) flows throughout the year and at low velocities off Baffin Bay (Huang et al., 2024; Curry et al., 2014). Rudels (2011) described



Transition Water as an intermediate layer formed by mixing Atlantic water from the south and colder waters from the north. More recently, Huang et al. (2024) identified two different types of Transition Water, each resulting from the mixing of several water masses. Below Transition Water, the water column is dominated by Baffin Bay Mode Water (BBMW), also known as Baffin Bay Deep Water, whose formation processes remain unclear (Huang et al., 2024; Bourke et al., 1989). In central Baffin Bay, depths greater than 1600 m are filled by the old Baffin Bay Bottom Water (Bourke et al., 1989), which does not outflow Davis Strait.

Inflowing into the Labrador Sea, the cold, fresh surface waters from the BIC join the Labrador Current, where they mix with waters from the WGC that turned westward, bathymetrically stirred at the Labrador Sea (García-Ibáñez et al., 2015; Cuny et al., 2002; Myers, 2005).

In the Labrador Sea, waters transported by the WGC may entrain eddies that detach from the boundary current and drift into the central basin. These waters also contribute to the formation of Labrador Sea Water (LSW, orange arrow in Fig. 1A), together with fresh waters from the Labrador Current, which account for approximately 6-8% (Myers, 2005; Lilly et al., 2003; Hátún et al., 2007). The annual deep winter convection that forms the LSW represents a key component of the Atlantic Meridional Ocean Circulation (Yashayaev, 2007; Bower et al., 2009; Lavender et al., 2005). Other water masses in the Labrador Sea include the North East Atlantic Deep Water (NEADW), which follows the cyclonic Deep Western Boundary Current and is found below LSW at approximately 2000 m. The NEADW forms by mixing of multiple source waters inflowing from east of Greenland, including LSW, Denmark Strait Overflow Water (DSOW, dark green in Fig. 1A), and Iceland Scotland Overflow Water (ISOW, light green in Fig. 1A) (Yashayaev, 2007; Tanhua, 2005; Dickson and Brown, 1994; García-Ibáñez et al., 2015). The bottom depths are occupied by DSOW carried primarily by the Deep Western Boundary Current from its formation region in the Nordic Seas towards the Grand Banks (Dale et al., 2024; Leist et al., 2024; Rudels, 2011).

Decades of oceanographic studies have significantly improved our understanding of water mass composition and volumetric transport in Davis Strait, Baffin Bay, and the Labrador Sea (Bourke et al., 1989; Yashayaev, 2007; Curry et al., 2011). However, climate-driven changes are altering the circulation and freshwater dynamics of the region (Shan et al., 2024). The Arctic Ocean is warming at nearly four times the global average under climate change, yet current ocean models appear to under-represent this rapid evolution (Rantanen et al., 2022). Observational evidence shows that weakened stratification in the Arctic Ocean is driving structural changes in the water column—a process known as atlantification. This shift enables warm Atlantic waters to penetrate further north, accelerating sea ice melt and reducing winter sea ice formation (Polyakov et al., 2025, 2017; Wang et al., 2024). Historically, approximately half of the Arctic freshwater has been exported through the Canadian Arctic Archipelago (CAA) and into Baffin Bay (Haine et al., 2015). In recent years, increased glacial melt from Greenland and the CAA (Rudels, 2011; The IMBIE Team, 2020; Gardner et al., 2011) has led to more frequent ice-free channels during summer (Canadian Ice Service; http://ice-glaces.ec.gc.ca/), likely modifying freshwater exchange between the Arctic and Baffin Bay (Malles et al., 2025). Furthermore, Baffin Bay now receives increasing volumes of Greenland glacier meltwater, a trend accelerated by the warming influence of Atlantic waters (Holland et al., 2008).

Established pathways of water mass exchange between high latitudes and the subpolar North Atlantic may also be changing




(Weijer et al., 2022; Carmack et al., 2016). Freshwater discharge (water with practical salinity below 34.6 PSU Zhang et al. (2021a)) from the Davis Strait into the subpolar North Atlantic is projected to increase, potentially affecting deep convection in the Labrador Sea (Zhang et al., 2021b; Wang et al., 2023; Yashayaev, 2024; Zhang et al., 2021a). Despite Davis Strait's critical role as a changing freshwater source to the subpolar North Atlantic (Shan et al., 2024), the freshwater dynamics of both Davis Strait and Baffin Bay remain poorly constrained. This is largely due to strong seasonal variability in freshwater release and the limited access during winter imposed by sea ice cover (Curry et al., 2014; Haine et al., 2015).

Despite the limitations in assessing the water mass formation in Baffin Bay, the contribution from the Labrador Sea to the Baffin Bay, via Davis Strait is well studied. However, the contribution of water outflowing Davis Strait on the water mass formation in the Labrador Sea remains uncertain. This is likely due to intense winter convection in the Labrador Sea, which strongly modifies temperature and salinity, complicating efforts to trace the origins of water using only these properties (Rudels, 2011; Yashayaev, 2007; Clarke and Gascard, 1983).

The role of Baffin Bay intermediate and deep waters (e.g. Transition Water) in the formation of LSW remains poorly understood compared to the better-known origin and pathways of surface waters (Curry et al., 2014; Cuny et al., 2005). Recent studies have emphasised the importance of Arctic freshwater export to the subpolar North Atlantic (Malles et al., 2025; Duyck et al., 2025; Myers, 2005) and the role of cross-density lateral mixing in the Labrador Sea, contributing approximately 60% to the annual formation of LSW (Zou et al., 2023; Pickart and Spall, 2007). However, most observational studies continue to focus on boundary current systems, leaving off-boundary water exchanges understudied (Huang et al., 2024; Curry et al., 2011; Azetsu-Scott et al., 2012), which could be crucial to understanding phytoplankton bloom formation and carbon cycling/sequestration (Boyd et al., 2019). Therefore, a better understanding of the off-shelf and intermediate circulation between Baffin Bay and the Labrador Sea is still needed (Rudels et al., 2004), particularly regarding the origin and formation of Transition Water and potential contributions of Arctic freshwater to the formation of LSW.

## 1.2 Using radionuclides to trace water mass origin

To advance on these complex water-mass interactions, we propose a novel approach in the region that combines two long-lived artificial radionuclides $^{129}$I (T$_{1/2}$=15.7 Ma) and $^{236}$U (T$_{1/2}$=23.4 Ma), with conservative hydrographic properties (salinity and temperature). We expect $^{129}$I and $^{236}$U to trace circulation patterns and mixing in the Davis Strait region because inflowing waters present contrasting tracer concentrations measured upstream in the Pacific, Atlantic and Arctic oceans (Wefing et al., 2021; Payne et al., 2024; Leist et al., 2024).

$^{129}$I and $^{236}$U originate primarily from nuclear sources and are assumed to be conservative in seawater(Christl et al., 2015; Casacuberta and Smith, 2023). Their input into the ocean was dominated by a peak in $^{236}$U from the global fallout from the weapon test in the 1960s, and the liquid releases from nuclear reprocessing plants of Sellafield (UK) and La Hague (F)(Fig.1A black factory symbols), peaking in the 1980s for $^{236}$U and ramping up after 1990s for $^{129}$I. Fig. A1 represents their input function defined at 70°N, a combination of global fallout and releases from the nuclear reprocessing plants. In both cases, the released radionuclides flow north from the North Sea and join the Norwegian Coastal Current (NCC), entering the Arctic Ocean. From the entrance of the Arctic, the radionuclides can recirculate within the Arctic Basin in a short loop (central Arctic)



or a longer path (Canada Basin) before exiting the Arctic via Fram Strait or the CAA (Li et al., 2020; Rudels et al., 1994; Payne et al., 2024; Casacuberta and Smith, 2023). From Fram Strait, the radionuclides are transported south and ultimately reach the Labrador Sea and Baffin Bay as part of the WGC (Wefing et al., 2019; Dale et al., 2024; Leist et al., 2024). Up to date, there are no measurements of these two tracers in either the Nares Strait or the CAA, so their transport through these routes is still unknown.

In the Arctic, the transport times and mixing of Atlantic Water can be calculated using the input function shown in Fig. A1 (Wefing et al., 2021; Payne et al., 2024). However, in sub-Arctic regions, the simultaneous mixing of multiple water masses with distinct input functions complicates the accurate estimation of water age and mixing when relying solely on these tracers. Nevertheless, the combination of $^{129}$I and $^{236}$U has already proved to be a suitable tracer of the formation and origin of the water mass in the subpolar North Atlantic, where it can be assumed that both tracers are in a steady state for each water mass (Castrillejo et al., 2018; Leist et al., 2024; Dale et al., 2024). This is a valid assumption, since recent changes (approximately 20 years) in the input function are small compared to changes caused by the mixing of different water masses, each carrying a distinct tracer signal.

In the study area, we expected to find a wide representation of the historical temporal variation of the tracers, allowing to disentangle the origin of Atlantic waters: While the "old" Arctic-Atlantic water from the Canada Basin would be characterised by high $^{236}$U and low $^{129}$I (Payne et al., 2024), the "young" Arctic-Atlantic water recirculated in the central Arctic would carry a higher signal of $^{129}$I (Wefing et al., 2025). Low tracer concentrations should be characteristic of the WGIW, which enters the study region from the subtropics without direct contact with reprocessing (Castrillejo et al., 2018; Dale et al., 2024; Leist et al., 2024). Also, Pacific Water (32.5 PSU) entering Baffin Bay through CAA would carry low concentrations of both radionuclides, as their only source was the atmospheric weapon tests (Payne et al., 2024). Finally, it is important to note that the freshest waters (i.e. glacier melt, sea ice melt, and river runoff) should contain almost no tracer, hereinafter denoted as tracer-free waters.

In this study, we address critical knowledge gaps in the understanding of water mass transformations and exchanges between Baffin Bay and the Labrador Sea by applying a novel approach based on the two long-lived artificial radionuclides, $^{129}$I and $^{236}$U. The tracers, which exhibit distinct distributions across Arctic and Atlantic origin water, offer a complementary tool to traditional hydrographic parameters, enabling the identification of water mass sources in regions characterised by complex mixing and strong seasonal variability. Using observations collected between 2022 and 2024 across key Arctic gateways —including Davis Strait, Nares Strait, Lancaster Sound— and the Labrador Sea, we apply a binary mixing model to characterise tracer signatures of inflowing water masses and quantify their contributions to key intermediate and deep water masses. This approach allows us to explore the origin of Transition Water, Baffin Bay Mode Water and Arctic Water and to follow the evolution of West Greenland Shelf Water and West Greenland Irminger Water. It further allows for the evaluation of the potential influence of Transition Water on water mass formation processes in the Labrador Sea.




## 2 Methods

### 2.1 Sample collection and processing

Seawater samples were collected from multiple sites (Fig.1B). In May 2022, seawater was collected from six depth profiles
along the AR7W Line across the Labrador Sea aboard R/V Atlantis as a part of the Atlantic Zone Off-Shelf Monitoring Program
(AZOMP) by Bedford Institute of Oceanography (red triangles). In October 2022, additional sampling was conducted in the
Davis Strait region aboard R/V Neil Armstrong as part of the Davis Strait Observing System (DSOS) Program. This included a
depth profile in central Baffin Bay (Fig.1, red square), and full transects along the Northern Line (9 stations, Fig. 1 red circles),
Davis Strait Mooring Line (10 stations, red diamonds Fig. 1B) and Northern Labrador Sea Line (9 stations, Fig. 1 red stars).
Further sampling was conducted in September-October 2024 during the Transforming Climate Action Legs 4 and 5 onboard
CCGS Amundsen, targeting Nares Strait (orange unfilled symbols) and Lancaster Sound ( red unfilled symbols, Fig. 1B). In
northern Nares Strait, one depth profile (50-345 m, St. NS89) was sampled along with a single-depth station (NS79). In the
southern Nares Strait, located south of the sill, four stations were sampled, NS102 to NS114, with one depth profile (70-600 m)
at NS107. In the Lancaster Sound, two sites were sampled in the western part within the Archipelago (KEBABB S3, 2-140 m),
and the eastern part neighbouring Baffin Bay (TCA S3, 2-890 m). All cruises included a CTD-Rosette equipped with 24-12 L
Niskin bottles, which were used to collect seawater.

The seawater samples for $^{129}$I analysis were collected in 250 ml opaque plastic bottles, and pre-rinsed three times with the
sample before filling. For $^{236}$U, 2-3 L of seawater were collected in Nalgene Cubitainers. All samples were stored unprocessed
and sent to ETH Zurich for the analysis of iodine and uranium isotopes.

The radiochemistry of $^{129}$I and $^{236}$U was performed following Wefing et al. (2019) and Leist et al. (2024) for a total of
307 samples. The samples were analysed using the TANDY AMS facility at the Laboratory of Ion Beam Physics (LIP), ETH
Zurich (Vockenhuber et al., 2015). Reproducibility was estimated based on repeated measurements of an internal seawater
standard (n=14, average= 10.4±0.5×10$^7$ at/kg). Blanks (n=16, 1.2±1.5×10$^7$ at/kg) were obtained using Milli-Q water treated
following the same procedure as for seawater samples.

The samples for $^{236}$U were spiked with $^{233}$U following the radiochemistry described by Wefing et al. (2021) and Leist et al.
(2024). The MILEA AMS facility at LIP (Christl et al., 2013, 2023) was used to measure $^{233}$U/$^{238}$U and $^{236}$U/$^{238}$U in the
samples and the in-house standard "ZUTRI". Correction procedures were applied as described by Christl et al. (2023). The
concentrations of $^{236}$U and $^{238}$U were calculated based on the known amount of $^{233}$U that was spiked in the sample. Each
batch included one blank (n=11, $^{236}$U:0.03±0.02×10$^7$ at/kg), consisting of MilliQ water that was treated like the samples.

The combined uncertainty of chemical processing and measurements for both $^{129}$I and $^{236}$U was below 6%. Detailed ana-
lytical uncertainties are reported in the Zenode database (https://doi.org/10.5281/zenodo.16914587).

### 2.2 Water mass classification

The water masses (summarised in Table A1) were assigned according to previous classifications of the water mass in the liter-
ature (Curry et al., 2014; Huang et al., 2024; García-Ibáñez et al., 2015; Bourke et al., 1989) and identified using conservative





temperature (CT) and absolute salinity ($S_A$, see the T-S diagram in Fig. A2). The conversion from practical to absolute salinity and from potential to conservative temperature was made after TEOS-10 (IOC and IAPSO, 2010). West Greenland Irminger

Water (WGIW, CT >4°C, $S_A$ >34.7) is the warmest and most saline water mass, confined at intermediate depth along the Greenland shelf (Fig.A4E-H). West Greenland Shelf Water (WGSW, CT: 4.8°C, $S_A$ <34.2) is fresher than the WGIW and is located at the surface along the Greenland shelf (Fig.A4A-D). Arctic Water (AW, 1.0> CT >-0.76°C, $S_A$:>32.9) is the freshest water, confined to Baffin Island and the Canadian Shelf. Here, the range of CT and $S_A$ considered differs significantly from Curry et al. (2014), and we applied a narrower range for Arctic Water and differentiated it from cold Arctic Water (cold AW, CT

>-0.81°C, 32.5< $S_A$ <33.7), which is colder and more saline. Transition Water (1.4< CT <1.8°C, $S_A$ :34.6±0.1) and its fresher variant, Transition Water mix ($S_A$: 34.0-34.5, CT: 1.5°C), is characteristic of Baffin Bay and Davis Strait. In Baffin Bay, at depths of 600-1000 m, Baffin Bay Mode Water (BBMW, 0.7< CT <1.2°C, $S_A$: 34.6 Huang et al. (2024)) was found overflowing Baffin Bay Bottom Water (BBBW, CT <0.4°C, $S_A$: 34.6 Bourke et al. (1989), Fig. A4). In the Labrador Sea (Fig.A4C,D,G,H), Labrador Sea Water (LSW, 3.8°C<CT<3.1°C, $S_A$:35.0 García-Ibáñez et al. (2018)) is located at intermediate depth. Below

LSW, North East Atlantic Deep Water (NEADW, 2.0< CT <3.3°C, $S_A$: 35.07±0.2 Yashayaev (2007)) is overlying Denmark Strait Overflow Water (DSOW, CT <1.3°C, $S_A$: 35±0.1 García-Ibáñez et al. (2018)). In Nares Strait and Lancaster Sound, the water masses were classified as Arctic Water CAA ($S_A$ <33, CT <0) and Arctic Atlantic Water (AAW CAA, $S_A$ >33, CT >-1). Samples that do not fall within the description of water masses are considered to be a mixture of the previous ones. Although all samples are represented in the results section, the discussion will focus on the samples that were assigned to one of the

water masses.

## 2.3 Binary Mixing Model

To quantify the origin and mixing of water masses, we used a binary mixing model that combines both $^{236}$U and $^{129}$I, as described by Leist et al. (2024) and Dale et al. (2024). The water masses in the study area are considered to be the result of the conservative mixing of different source waters, referred to as endmembers, which establish the boundary tracer concentrations

and set the model domain. In this tracer space, mixing between endmembers is represented by mixing lines. The mixing fraction (fraction = length A/B) is calculated as the relative distance from a sample to its respective endmembers (A and B). The estimated fractions are calculated with the mean of the endmembers and the samples, and rounded values are reported. The specific endmembers used in this study are further discussed in Section 4.1.

One limitation of this model is that it considers mixing only along a single line between two endmembers. If a sample falls off

this mixing line, it needs to be orthogonally projected onto the line, if not stated otherwise. In addition, endmembers are derived from different numbers of samples (see Table A2), which might influence the error range. Therefore, fractions are calculated for the endmember means. The model also assumes a steady state for transient tracers, which was previously discussed and justified by Leist et al. (2024) and Dale et al. (2024). In the study region, changes in water mass formation and mixing ratios between endmembers are believed to have a greater influence on tracer concentrations than recent changes in the tracer input

function, supporting the steady-state assumption.



## 3 Results

This section presents individual depth profiles from Nares Strait and Lancaster Sound, followed by a detailed tracer distribution along the three transects at southern Baffin Bay (Northern Line), Davis Strait (Mooring Line), Northern and southern Labrador Sea (Northern Labrador Sea and AR7W lines). The results of $^{129}$I and $^{236}$U are reported and plotted in atoms per kg of seawater

(at/kg). To better reflect the general distribution patterns, each water mass is described across the lines/panels as a whole, rather than within individual sections. Details about tracer concentrations, uncertainties, and hydrographic properties are provided in the Zenodo database and supporting materials. The acronyms for the water masses can be found in the Appendix Table A1.

### 3.1 $^{129}$I and $^{236}$U in Nares Strait and Lancaster Sound

Depth profiles collected in Nares Strait are represented in Fig. 2A and B as orange symbols. Both tracers show low concentrations at the surface Arctic Water (AW, $^{129}$I: 60-85×10$^7$ at/kg, north $^{236}$U: 14×10$^6$ at/kg, south $^{236}$U: 11×10$^6$ at/kg) and a local maximum in the Arctic Atlantic Water at about 250 m (AAW, $^{129}$I: 180-210×10$^7$ at/kg, $^{236}$U:23×10$^6$ at/kg). Below that depth, $^{129}$I seems to slightly decrease towards the bottom, while $^{236}$U remains constant. Furthermore, the concentrations of

both isotopes appear to be higher in the northern part of the Nares Strait (squares) compared to the south (circles).

The two profiles in the Lancaster Sound are represented in Fig. 2A and B as red symbols. Similar to Nares Strait, the highest concentrations of $^{129}$I were associated with AAW located at 200 m in eastern Lancaster Sound ($^{129}$I: 130×10$^7$ at/kg, S$_A$: 33.85, CT: -0.10) and at depths below 140 m in the western station ($^{129}$I: 120×10$^7$ at/kg, S$_A$: 33.6, CT: -0.44). In western Lancaster Sound, the highest concentration of $^{236}$U was observed at 10 m depth in the Arctic Water ($^{236}$U: 19×10$^6$ at/kg,

S$_A$: 30.58, CT: -0.04), and the lowest at the surface ($^{236}$U: 15×10$^6$ at/kg). A similar distribution was observed in the eastern Lancaster Sound, with the lowest concentration at the surface ($^{236}$U: 13×10$^6$ at/kg), a maximum at 200 m ($^{236}$U: 22×10$^6$ at/kg), which then slightly decreased with increasing depth to $^{236}$U: 15×10$^6$ at/kg at 886 m.

### 3.2 $^{129}$I and $^{236}$U in central and southern Baffin Bay, Davis Strait and the Labrador Sea

Along all lines, the highest $^{129}$I concentrations were observed in West Greenland Shelf Water (WGSW) at the surface of the

Greenland shelf (Fig. 3A-D). $^{129}$I concentrations decreased along the pathway of the West Greenland Current (WGC), with WGSW dropping from 320×10$^7$at/kg at AR7W in Labrador Sea (Fig. 3D) to 170×10$^7$ at/kg at the Northern Line in Baffin Bay (Fig. 3A). In contrast, the warmest and most saline West Greenland Irminger Water (WGIW) generally showed increasing $^{129}$I concentrations from 50×10$^7$ at/kg at AR7W (Fig. 3D) to a maximum of 90×10$^7$ at/kg at the Northern Line (Fig. 3A). In AW, the freshest water along the Baffin Island Current (BIC) and Labrador Current (LC), $^{129}$I concentrations increased slightly

from central Baffin Bay (140×10$^7$ at/kg, blue squares Fig. 2C) to the Northern Labrador Sea Line (150×10$^7$ at/kg, Fig. 3C), but showed lower concentrations (100×10$^7$ at/kg) at the AR7W in the southern Labrador Sea. Cold Arctic Water exhibited elevated $^{129}$I concentrations in the 100-170×10$^7$ at/kg range and showed no clear north-south trend. While AW, WGIW and



WGSW were present in all sections, Transition Water was only present along the Mooring Line, Northern Line (Fig. 3B, A) and central Baffin Bay (Fig. 2C, blue squares), with concentrations in the range of 50-120$\times 10^7$ at/kg, respectively. $^{129}$I

concentrations at the levels of the blanks (<1.2$\pm$1.5$\times 10^7$ at/kg) were measured in BBBW present below 1600 m in central Baffin Bay and the Northern Line. Labrador Sea Water (LSW) and North East Atlantic Deep Water (NEADW) (only sampled at AR7W, Fig. 3D) covered an $^{129}$I range of 30-70$\times 10^7$ at/kg at intermediate depth in the Labrador Sea, while the near-bottom Denmark Strait Overflow Water (DSOW) presented concentrations up to 120$\times 10^7$ at/kg.

The distribution of $^{236}$U concentrations (Fig. 3E-H) differs from that of $^{129}$I. The highest $^{236}$U was observed in waters out-

flowing through Davis Strait (Fig. 3H, F), particularly with up to 19$\times 10^6$ at/kg in central Baffin Bay (Fig. 2D, blue squares), followed by cold AW, with $^{236}$U in the 14-17$\times 10^6$ at/kg range between the Mooring Line (Fig. 3F) and central Baffin Bay. While the $^{236}$U concentration in Arctic Water was generally in the range of 13-15$\times 10^6$ at/kg, the AR7W Line recorded the maximum value (16$\times 10^6$ at/kg, Fig. 3H). In water inflowing Davis Strait, such as WGSW, $^{236}$U decreased slightly towards the north, from 15$\times 10^6$ at/kg at the AR7W Line to 13$\times 10^6$ at/kg at the Northern Line (Fig. 3E). Similarly to $^{129}$I, low $^{236}$U

concentrations were measured in WGIW, with an increasing trend from 9$\times 10^6$ at/kg at AR7W to 11$\times 10^6$ at/kg on the Northern Line.

In the Labrador Sea, LSW presented $^{236}$U concentrations on the 9-11$\times 10^6$ at/kg range on AR7W and Northern Labrador Sea lines (Fig. 3H, G). NEADW, only sampled along AR7W, showed again $^{236}$U concentrations slightly above LSW (10-12$\times 10^6$ at/kg). The sampling density of DSOW was smaller in the Northern Labrador Sea Line than for AR7W and confined at slightly

lower concentrations (AR7W: 12-14$\times 10^6$ at/kg, Northern Labrador Sea Line: 12$\times 10^6$ at/kg). Finally, in BBBW occupying depths below 1600 m in central Baffin Bay (blue squares Fig. 2D) and the Northern Line, $^{236}$U concentrations reached the detection limit (1.2$\pm$1.5$\times 10^6$ at/kg).

Both tracers have concentrations well above the global fallout levels, which would be of 2.2$\times 10^7$ at/kg for $^{129}$I and 5.4$\times 10^6$ at/kg for $^{236}$U according to recent estimates at the Bering Strait (Payne et al., in prep). Only the bottom water at Baffin Bay

(BBBW) presents tracer concentrations lower than expected from the global fallout.

## 3.3   Water masses in T-S and $^{129}$I - $^{236}$U tracer space

Figure 4 provides an overview of the distribution of the water masses in T-S space (Fig. 4A) and $^{129}$I - $^{236}$U tracer space (Fig. 4B). In addition to the water mass description in Section 2.2, here the water masses have been further defined by looking at the radionuclide tracers. For example, in some cases, the $^{129}$I and $^{236}$U concentrations are used to define the same water

mass that changed the properties of temperature and salinity, but the tracer concentration remained the same (for example, WGSW in the AR7W Line falls within the temperature and salinity range of Arctic Water, Fig. 4A and B indicated by black cirles) as the tracer concentration results almost exclusively from mixing between endmembers. This is important to keep in mind in an area strongly affected by seasonality (Curry et al., 2014; Shan et al., 2024). The sampling of AR7W took place in spring, while the Northern Labrador Sea, Mooring and Northern lines, as well as the central Baffin Bay, were sampled in

autumn. WGSW (dark red symbols) is prominent in both graphs due to its high temperature and low salinity (Fig. 4 A), and because it contains the highest $^{129}$I (from 180 to >300 $\times 10^7$ at/kg) (Fig. 4B). Although seasonal variability is represented in





the broad temperature range (Fig. 4A) and is observed by, e.g. Curry et al. (2014) and Zweng and Münchow (2006), the tracer space shows a well-defined line, suggesting that the tracers are insensitive to seasonal changes.

Transition Water (dark blue symbols), located between cold Arctic Water (Fig. 4 teal symbols with golden edges) and WGI-
W/LSW (mint/orange symbols) in T-S space, shows the highest $^{236}$U (Fig. 4B), second only to concentrations measured at Nares Strait and Lancaster Sound. Lower tracer levels in the same region correspond to fresher surface waters.

Cold Arctic Water, characterised by its distinct temperature minimum and confined to the upper 100 m, is positioned in the tracer space at the mixing interface between the relatively warm Transition Water mix and WGSW, with a clear influence from low-tracer Pacific Water or water of subtropical North Atlantic origin, such as WGIW.

The saline and warm WGIW is located at the lower left in tracer space due to its low $^{129}$I and $^{236}$U concentration. Radionuclide tracers are especially low for samples collected in the Labrador Sea, then they increase slightly together with a freshening and cooling experienced on the northward flow of WGIW.

BBBW (purple squares and circles) in the Northern Line and central Baffin Bay is found in the lower left corner of the tracer space because its concentrations of $^{129}$I and $^{236}$U are extremely low and approach the analytical limit of detection. Both $^{129}$I and $^{236}$U increase in BBMW (light purple diamonds), with concentrations similar to the more saline and warmer LSW and NEADW in the Labrador Sea.

In the Labrador Sea, surface waters (red symbols) fill the tracer concentrations between WGSW and WGIW, as well as the salinity ranges between the two water masses. Two surface samples at AR7W are well separated from the remaining surface samples. They are colder (CT around 1.5°C) and with fresher (S$_A$:33.8) than the other surface samples (S$_A$>34.2). Their tracer concentration in $^{236}$U (more clearly seen in Fig. 6D), is elevated relative to the other surface samples and it falls within the cluster of fresher and colder DSOW.

DSOW (Fig. 4, green symbols), in turn, is characterised by low temperatures, high salinity, and intermediate tracer concentrations.

Freshwater sources - such as sea ice and glacial meltwater, river run-off, and precipitation - are considered tracer-free; thus, their presence would be indicated by lower salinities and a spread toward low tracer concentration.

## 4  Discussion

To investigate the origin and mixing of the water masses of this study, we apply the mixing model described in Section 2.3. Here, we first examine the endmembers used in the mixing model (Section 4.1), followed by a discussion of the evolution of West Greenland Shelf Water and West Greenland Irminger Water (WGSW and WGIW) as they enter Baffin Bay (Section 4.2). The origin of Transition Water, Baffin Bay Mode Water (BBMW), cold Arctic Water and Arctic Water is then examined in Sections 4.3 and 4.4. The discussion concludes with how all these waters contribute to the formation of Labrador Sea Water (LSW) and North East Atlantic Deep Water (NEADW, Section 4.5).



## 4.1 Endmembers in the mixing model

Endmembers will be used in the binary mixing model to estimate the origin and composition of water masses. In Fig. 5A, the
geographical location of the defined endmembers are shown as diamonds, and unfilled symbols, which are then represented in
Fig. 5B with their $^{129}$I and $^{236}$U tracer signatures.

These endmembers (Table A2) are defined using a combination of published data and new tracer measurements. The water
mass assignments are based on hydrographic properties, as explained in Section 3. For endmembers derived from the literature,
the original classification of the water mass is maintained.


### 4.1.1 Endmembers in the subpolar North Atlantic and the Canada Basin

The NAC endmember (turquoise diamond, Fig. 5) represents the North Atlantic Current, which feeds into the WGIW. It carries
water of mainly subtropical Atlantic origin labelled with a low tracer content from global fallout and a small contribution
from the European nuclear reprocessing plants (Leist et al., 2024; Castrillejo et al., 2018). Representative of this endmember
is a sample taken east of Reykjanes Ridge (St. 38, SAIW, Castrillejo et al., 2018). The diluted repocessing plant signal may
originate from tracer-labelled water entering the subpolar gyre from the Labrador and Irminger seas and reaching east of the
Reykjanes Ridge. The Iceland Scotland Overflow Water (ISOW), formed by deep convection north of Iceland, is represented
by data collected before entering the Icelandic Basin on the eastern side of the Reykjanes Ridge (Dale et al., 2024).

The Polar Surface Water - East Greenland Current endmember (PSW-EGC, dark red diamond, Fig. 5) represents the outflow
surface waters in the Arctic Ocean, which are then transported south by the East Greenland Current (Fig. 1A). This endmember
corresponds to a sample collected near the Denmark Strait in 2021 (St. MG17, Dale et al., 2024) and represents surface waters
entering the Labrador Sea via the West Greenland Current (WGC). As described in Dale et al. (2024) and Leist et al. (2024), this
sample is the most characteristic of PSW-EGC in southern Greenland, still containing a significant fraction of PSW (Dale et al.,
2024) that outflows the Arctic Ocean (Wefing et al., 2025). The PSW-EGC endmember falls within a dilution line between
NAC and the PSW sampled in the Central Arctic in 2021 by Wefing et al. (2025) (grey circles, Fig. 5). The PSW-EGC and
PSW from the central Arctic both represent "younger" water imprinted with post-1980s reprocessing releases (Wefing et al.,
2025) with high $^{129}$I and comparably low $^{236}$U (Fig. A1).

In contrast, the "old" Arctic Atlantic Water (AAW) in the Arctic Canada Basin at isopycnal $\sigma_\theta$= 27.93 (AAW CB, grey crosses,
Fig. 5), contains high $^{236}$U and comparably low $^{129}$I (St. MK2, MK3 and CB29 in Payne et al., 2024) from global fallout
and reprocessing discharges prior to the 1990s (Fig. A1). Above AAW in the Canada Basin, the Pacific Summer Water holds
especially low $^{129}$I and $^{236}$U from global fallout alone (dark green diamond in Fig. 5, all stations of JOIS 2020 Payne et al.
(2024)). We also include an endmember called Tracer free, which consists of sea ice and glacial meltwater, precipitation, and
river runoff.

Fresh water, such as precipitation, river runoff, sea ice melt and glacial meltwater, is referred to as Tracer-free and indicated as
a diamond with a black outline in Fig. 5B. The Pacific Water, low in salinity, has only elevated $^{236}$U from the global fallout,





while the WGIW, even though its hydrographic properties are very different, has low tracer concentrations as well. Therefore, it is not possible to disentangle the different contributions of these water masses. In future work, this might be addressed by adding other tracers such as $\delta^{18}$O.

### 4.1.2 Endmembers in Nares Strait and Lancaster Sound

The measurements in Nares Strait and Lancaster Sound presented in this study provide suitable new endmembers for the northern exchange via these channels. They are represented in Fig. 5 as orange (Nares Strait) and red outlined symbols (Lancaster Sound). High tracer concentration observed below 200 m in northern Nares Strait might reflect the mixture between the Arctic-Atlantic layer from the Canadian Basin (grey diamond) and the central Arctic (grey circle). On the contrary, low $^{129}$I and $^{236}$U at shallower depths may be due to inflowing Pacific Water. In southern Nares Strait, the observed tracer concentrations can be

explained by mixing AAW from northern Nares Strait with about 25% of Pacific Water or Tracer-free water.
Samples from Lancaster Sound (red outlined symbols in Fig. 5) have generally lower tracer content than in Nares Strait. However, most Lancaster Sound samples appear to reflect a mixture of Arctic Atlantic Water in the Canadian Basin (50–60%) with freshwater from the Pacific or Tracer-free water.

### 4.2 Evolution of West Greenland Shelf Water and West Greenland Irminger Water

West Greenland Shelf Water (WGSW) and West Greenland Irminger Water (WGIW) undergo strong seasonal hydrographic variability, with cooling and salinification in winter and refreshing in summer (Curry et al., 2014), which complicates quantifying their role in water mass formation in the Baffin Bay (Curry et al., 2014). In contrast, radionuclide tracers remain unaffected by these processes, making them a powerful tool for tracking WGSW and WGIW in their journey to northern latitudes. In our dataset, the largest hydrographic variations are observed in WGSW: in spring samples from the AR7W Line show low

temperatures (dark red triangles in Fig. 4A), while autumn samples adjacent to Davis Strait display higher temperatures and a broad salinity range (dark red symbols in Fig. 4A). Despite these seasonal changes, tracer concentrations (Fig. 4B) remain remarkably stable, allowing us to robustly quantify the contribution of the WGSW to Arctic Water formation.
Using the binary mixing model, we find that the WGSW in the AR7W Line (Fig. 6A as dark red triangles) retains up to 70% of the PSW-EGC measured at Denmark Strait (fraction PSW-EGC = A/B, Fig. 6A, see Section 2.3 for fraction calculation). As

the WGSW flows northwards along the Greenland shelf, it becomes progressively entrained by tracer-free waters and WGIW (light blue symbols in Fig. 6A). By the time WGSW reaches the Mooring and Northern Line (Fig. 6A diamonds and circles), the PSW-EGC fraction has decreased to about 40%, consistent with estimates by Huang et al. (2024) for southern Baffin Bay (160 m out of 600 m depth) and what Münchow et al. (2015) referred to as "anomalous waters" in the upper 800 m at the Greenland slope. At central Baffin Bay, water classified as Arctic Water (teal square, Fig. 6A), preserved a significant PSW-

EGC fraction of about 30%. This was estimated as the fraction PSW-EGC = C/B, with the Arctic Water sample projected to the PSW-EGC-NAC mixing line in an extension of mixing with Transition Water. This finding contrasts with Huang et al. (2024), who did not consider WGSW as a major contributor to the formation of water masses in the region. The substantial presence of WGSW may underscore the role of eddies, the off-branching character of the WGC, and the significant transformation of





surface waters in Baffin Bay.

WGIW (Fig. 6A light green symbols) lies closer to the NAC endmember than WGSW, because these waters have a stronger influence from waters coming directly from the south and thus have less tracer content. An increase in its tracer content from the Labrador Sea to Baffin Bay confirms its mixing with WGSW, preserving approximately 15% PSW-EGC (fraction PSW-EGC = 1-D/B), consistent with Huang et al. (2024), while contrasting Rysgaard et al. (2020), who did not observe WGIW on the Greenland shelf in Baffin Bay. Furthermore, Huang et al. (2024) observed WGIW (200 m of 500 m) in central Baffin Bay,

which could not be identified here, probably due to the limitation of the mixing to two endmembers in the tracer analysis.

## 4.3 Origin of Transition Water and Baffin Bay Bottom Water

The formation and origin of Transition Water is a topic of ongoing debate. While Huang et al. (2024) described Transition Water (their TrW1) as a mixture of several water masses (WGIW, cold Arctic Water, and WGSW), Rudels (2011) referred to a blend of "Atlantic" water from the south and denser, colder northern water. In T-S space (Fig. 4A), this formation is well

supported by a mixture of cold Arctic Water with WGIW. However, the notably elevated concentrations of $^{236}$U and low $^{129}$I in Transition Water along the Northern Line (circles in Fig. 7A), Mooring Line (diamonds) and central Baffin Bay (squares) cannot be explained by mixing low-$^{236}$U WGIW with intermediate-$^{236}$U cold Arctic Water or WGSW with high concentrations of $^{129}$I. This suggests that another endmember with more elevated $^{236}$U must contribute to the formation of Transition Water. The only waters with such a high $^{236}$U content are found in Nares Strait and Lancaster Sound (Fig. 7B). The AAW outflowing

Lancaster Sound meets this condition best when considering also the temperature and salinity of the Transition Water.

To examine this further, we revisit Transition Water in $^{129}$I-$^{236}$U space (dark blue symbols in Fig. 6A). A clear similarity emerges between Transition Water and waters from Lancaster Sound (red symbols in Fig. 6A), particularly those from its western part. These waters likely result from the dilution of AAW in the Canadian Basin by fresher ($S_A$<32), low-tracer waters derived from the Pacific Ocean inflow, river runoff, and sea ice meltwater. Our observation of AAW outflow from the Canadian

Basin contrasts with most studies, which have typically considered Pacific-origin water as the main component of Arctic waters in the Baffin Bay (Goosse et al., 1997; Jones et al., 2003). None of these studies discusses the presence of AAW making its way through the Canadian Archipelago. This raises an important question: how can waters of Arctic-Atlantic origin, typically found below 200 m in the central Arctic Ocean, pass through the relatively shallow channels of the CAA? The estimated 40–45% contribution of AAW from the Canadian Basin to Transition Water formation (fraction AAW at CB = E/F, accounting for

mixing with Pacific Water) and its presence in Lancaster Sound, is likely the result of upwelling in the Canada Basin. Easterly wind events, associated with the Beaufort High or synoptic low-pressure systems originating in the North Pacific, can result in the upwelling AAW onto the shallow Chukchi and Beaufort Sea shelves, where AAW can be transported towards the Canadian Arctic Archipelago as part of the Beaufort Sea shelfbreak jet (Pickart et al., 2013; Lin et al., 2019; Yang, 2006; Pickart et al., 2009). While storms associated with AAW upwelling are centred over the Allutian Islands in the Bering Sea (Pickart et al.,

2009), as these storms travel northward into the Beaufort Sea, they can cause cyclonic wind stress anomalies over the Canada Basin, which in turn, increase volume transports through Lancaster Sound (Zhang et al., 2016; Peterson et al., 2012). These processes can drive the transport of denser shelf waters containing AAW southward along the deeper passages of the CAA,



such as M'Clintock Channel, eventually connecting to Lancaster Sound and northern Baffin Bay (Wang et al., 2012). Once these mixed Pacific-AAW shelf waters reach northwestern Baffin Bay, they can be transported from shallow to intermediate depths by deep convection associated with large air-sea fluxes and polynya development (Wang et al., 2012; Vincent, 2019; Yao and Tang, 2003; Aagaard and Carmack, 1989), and follow the strong lateral exchange between the slope and basin as modelled by e.g. Münchow et al. (2015). From intermediate depths in northern Baffin Bay, particularly at the 27.5 kg/m$^3$ isopycnal ($\sim$ 300 meters Lobb et al. (2003)), these dense waters can encounter warm WGIW, where diffusive instability and cabbeling can cause these waters to further mix and descend in the water column (Lobb et al., 2003; Shan et al., 2024; Huang et al., 2024). In summary, while there is substantial physical uncertainty regarding the presence of AAW in the CAA, these processes provide a plausible mechanism for the high $^{236}$U concentrations within the watermass in central and southern Baffin Bay. The formation of Transition Water from only two main water masses (AAW, WGIW; Fig. 6A) contrasts with the interpretation by Huang et al. (2024) involving more water bodies. Further, by tracing its origin to Lancaster Sound and the Canadian Basin, our findings add further detail to the broader framework proposed by Rudels (2011), in which Atlantic water of southern origin and dense northern water mix. Results further show that local hydrographic modifications described in literature (Curry et al., 2014; Tang et al., 2004; Shan et al., 2024; Melling et al., 2001) have no impact on the $^{129}$I and $^{236}$U fingerprint. In the calculated AAW fraction, only mixing between AAW in the Canada Basin and Pacific Water is taken into account, because the single contributions of WGIW vs. Pacific Water vs. Tracer-free Water can not be separated. Here, the model is reaching its limitation since the water masses have low tracer concentrations, and the ratio between the tracers is too similar. The NAC endmember even aligns with the mixing between AAW from Canada Basin and Pacific Water. If one considers a main dilution by WGIW, our estimate of the AAW fraction may be in the lower range.

Similar to Transition Water, the origin and formation of BBMW (purple symbols in Fig. 6A) remains unclear (Huang et al., 2024). BBMW has been suggested to be older than other water masses in the region, with tracer ages estimated to be up to 90 years (Lique et al., 2010). The relatively high $^{236}$U and very low $^{129}$I displayed by BBMW may again indicate the presence of AAW transported from the Canadian Basin, and with the imprint of "old" waters, according to Fig. A1. These ages would agree with older AAW outflowing through the CAA, labelled with relatively elevated $^{236}$U and almost no $^{129}$I.

### 4.4 Origin and formation of Arctic Water, cold Arctic Water and Transition Water mix

Arctic Water and cold Arctic Water are significant sources of freshwater to the subpolar North Atlantic, as they are strongly influenced by sea ice and glacial meltwater (Huang et al., 2024; Curry et al., 2014; Shan et al., 2024). However, the composition of these freshwaters as they mix with other water masses in the region still remains under debate. The Radionuclide tracers $^{236}$U and $^{129}$I demonstrate that cold Arctic Water is a mixture of Arctic Atlantic Water outflowing mostly through the Nares Strait, Pacific Waters outflowing through the Canadian Archipelago and WGSW coming from southern latitudes. Arctic Water has a similar origin, but with a stronger influence of tracer-free freshwater. In addition, the tracers reveal a distinct water mass, here termed Transition Water mix, which has previously been grouped under Transition Water but appears to originate differently. Arctic Water (teal symbols) and cold Arctic Water (teal symbols with golden edges), as well as Transition Water mix (light





blue), are shown in Fig. 6B, alongside new endmembers, derived from the mean values of Transition Water and WGSW, in Fig. 6A. Error bars indicate the standard deviation of the Transition Water and WGSW data points presented in the previous panel. For WGSW, the mean and standard deviation were derived for $^{129}$I concentration below $240\times10^7$ at/kg to best represent

the core of WGSW reaching north of Davis Strait. Cold Arctic Water has generally higher tracer concentration than Arctic Water and is confined to a cold (CT about -1.5 °C) and rather narrow temperature range, while it covers a salinity range between 32.5 and 33.6 (Fig. 4 and A2). Although Huang et al. (2024) observed mixing between cold Polar Water (similar to this study cold Arctic Water), Meteoric Water, as well as WGIW, the mixing model here (Fig. 6B) points to additional waters contributing to its formation. In the following, we examine each of them in detail, starting with WGSW.

In the binary mixing plot, cold Arctic Water (Fig. 6B, teal symbols with golden edges) stretches from the mixing line of Nares Strait South (NS South) and Pacific Water towards the endmember of WGSW, suggesting the influence of WGSW on cold Arctic Water as well. The largest contribution of WGSW, approximately 65%, is observed in a sample at a depth of 54 m in the middle of the Northern Line as indicated in Fig. 6B) by an "a", and the fraction WGSW = G/H. WGSW generally follows the main cyclonic circulation (Curry et al., 2011; Azetsu-Scott et al., 2012; Huang et al., 2024) and may cool through air-sea heat

exchange, especially in the North Water polynia in winter, reaching the temperature of the cold Arctic Water (Yao and Tang, 2003).

AAW (fraction AAW=I/J), probably outflowing through the Nares Strait, contributes up to 35% to cold Arctic Water ("b" in Fig. 6B). The role of the outflow of the Nares Strait, previously observed by Huang et al. (2024) and Melling et al. (2001), is now reinforced based on a completely different set of tracers.

Pacific Water and WGIW, both characterised by low $^{129}$I and $^{236}$U may contribute to cold Arctic Water. While Huang et al. (2024) focused primarily on WGIW, the tracer data is less conclusive, as mixing with either source produces a similar low-tracer signal ("c" in Fig. 6B). At this stage, the binary mixing model reaches its limit, unable to distinguish between these two low-tracer endmembers.

Finally, mixing with Arctic Water, as previously noted by Huang et al. (2024), is likely but cannot be precisely quantified here

due to similar $^{129}$I and $^{236}$U signatures of cold Arctic Water and Arctic Water. Overall, the cold Arctic Water cluster reflects a varying blend of water masses described above, showing no clear trend in hydrographic properties.

Following the assessment of the cold Arctic Water cluster, we turn to Arctic Water to explore its distinct properties and its close association with cold Arctic Water in tracer space. Although Arctic Water is fresher than cold Arctic Water, both water masses plot closely together in tracer space (teal in Fig. 6B). Arctic Water appears to be the result of the dilution of cold Arctic

Water with low-tracer waters, probably of Tracer-free and Pacific origin (Huang et al., 2024; Planat et al., 2025). Unlike cold Arctic Water, Arctic Water shows the strongest WGSW influence in central Baffin Bay, highlighting the strong stratification in this region. The input of low tracer water, similar to that in cold Arctic Water, is most pronounced along the western side of the Northern Line ("d" in Fig. 6B), confined within the BIC. Stronger mixing with cold Arctic Water likely occurs near the Mooring Line, illustrating the dynamic conditions in this area (Huang et al., 2024; Tang et al., 2004). Further south in

the Northern Labrador Sea Line (stars), Arctic Water was likely sampled within the bifurcation of the WGC, showing up to 30% WGSW ("e" in Fig. 6B, fraction WGSW = K/L, "e" projected onto line "L" parallel to line "H"). In comparison, at the





AR7W Line (triangles), Arctic Water carried by the Labrador Current primarily transports the AAW tracer signal with up
to 30% contribution when mixing of NS South with Pacific Water is considered ("f" in Fig. 6B, fraction AAW=M/J). The
transition from a predominantly WGSW endmember to AAW — potentially sourced from Nares Strait— remains unresolved
based on the current data. Possible influences on the main water masses observed in the Labrador Current might be caused by
the seasonal variability in the southward velocities observed at Davis Strait, which are generally highest in summer and lowest
in winter (Curry et al., 2014; Myers, 2005; Shan et al., 2024) or the outflow of AAW (CB) via Hudson Strait, south of the
Northern Labrador Sea Line (Straneo and Saucier, 2008). Water classified as Transition Water mix bridges Transition Water
and cold Arctic Water in TS and tracer space (light blue in Fig. 4 and 6B) and may represent the mixing and cooling of AAW
from Lancaster Sound and/or Nares Strait and WGSW. Here, the binary mixing model is less conclusive about the origin of
AAW, since mixing with WGSW would spread AAW from Lancaster Sound towards the ratios observed for AAW outflowing
Nares Strait. Therefore, AAW originating from Lancaster Sound and Nares Strait needs to be considered.

### 4.5   Implications on the formation of Labrador Sea Water and North East Atlantic Deep Water

The formation of Labrador Sea Water (LSW) is governed by complex physical processes (Clarke and Gascard, 1983) and
influenced by various freshwater sources (salinity <34.6 PSU, Zhang et al. (2021a)) that can impact its properties (Yashayaev,
2007; Yamamoto-Kawai et al., 2008). To date, the relative contributions of these freshwater sources remain unresolved, with
most studies focusing on boundary inputs (Schmidt and Send, 2007; Zhang et al., 2021a). In contrast, artificial radionuclides
reveal the potential significance of off-boundary sources, such as Transition Water and Baffin Bay Mode Water, not only for
LSW formation but also for the North East Atlantic Deep Water (NEADW).
In the previous study by Leist et al. (2024) using $^{129}$I and $^{236}$U at the Labrador Sea region, the mixing model reached its limi-
tations when trying to understand the origin of LSW, due to low tracer concentrations and the absence of critical endmembers.
However, the study identified $^{129}$I rich WGSW eddies as one of the sources contributing to LSW as they branch off the WGC
and into the interior of the Labrador Sea (Leist et al., 2024; Hátún et al., 2007; Holliday et al., 2009; Lilly et al., 2003; Chanut
et al., 2008; Pacini and Pickart, 2022). Labrador Sea Water (red symbols in Fig. 6C) appear to have a significant contribution
of WGSW, estimated here to be up to 20% (WGSW fraction = N/L). The influence on surface waters in the Labrador Sea
(dark orange in Fig. 6C) is even higher, up to 40% (WGSW fraction = O/L, crossing point of surface sample with mixing line
estimated along the linear regression between all surface samples).
Another water mass that might contribute to the tracer load in LSW is DSOW, as previously discussed in Dale et al. (2024) and
Leist et al. (2024). The formation of DSOW north of the Denmark Strait is complex and discussed in greater detail by Dale
et al. (2024) and Tanhua et al. (2005). One possible source of DSOW is a branch of the EGC carrying water of Arctic origin
outflowing via Fram Strait, which ultimately contributes to the deep overflow at the Denmark Strait. During this convection
process, it can already mix with LSW, present in the Irminger Sea, and circulate within the subpolar gyre (Lavender et al.,
2005; Zou et al., 2023). Once at the Labrador Sea, DSOW is confined to the bottom of the basin and separated from LSW by
NEADW. However, tracer similarities between surface samples and DSOW were still observed in the Labrador Sea in 2022.
See for example the two AR7W surface samples in Fig. 6C (highlighted by "g"), which fall within the DSOW cluster, despite





differences in their temperature and salinity properties (Fig. 4, colder (CT <1.5°C) and fresher ($S_A$ <34) compared to the other surface samples) and located close to the WGC at the AR7W Line.

Another source of $^{129}$I and $^{236}$U in the LSW could be the Labrador Current, carrying Arctic Water and cold Arctic Water, but this would be relatively small (< 10%), as already discussed in Myers (2005); Pickart and Spall (2007); Wang et al. (2018);

Duyck et al. (2025). However, the main LSW cluster observed in Fig. 6C (red triangles) appears to have shifted to the left of the mixing line between the DSOW cluster and NAC. This suggests that another water mass, richer in $^{236}$U relative to $^{129}$I, is required to account for the observed composition of LSW. This feature, first noted in 2014 by Castrillejo et al. (2018) and later confirmed by Leist et al. (2024), can now be explained by the influence of Transition Water, which flows out of Davis Strait and entrains into LSW. Among the water masses entering the Labrador Sea, Transition Water is the only one with sufficiently high

$^{236}$U concentrations to explain the tracer signature of LSW. To our knowledge, this entrainment of Transition Water into LSW has not been previously considered in the literature. As shown by "h" in Fig. 6C, this contribution could be as high as 30% (fraction TrW = P/Q). In Davis Strait, long-term moored measurements of temperature and salinity, and across-strait velocity from 2004-2024 at 60°W and 500 meters at the Mooring Line support this argument and show that Transition Water flows southward from Davis Strait year-round into the Labrador Sea (Fig. A3), complementing gridded transports for the 2004-2010

period (Curry et al., 2014). These measurements are located near the $^{236}$U maximum in the Mooring Line ($^{236}$U$\sim$ 17.5 $10^6$ at/kg) and demonstrate the consistent interannually southward transport of waters fresher than LSW ($S_A < 34.9$ g/kg) into the Labrador Sea, providing both a $^{236}$U and freshwater source for LSW. Once Transition Water enters the Labrador Sea, it may become entrained with LSW throughout the year through cross-density mixing, as observed by Zou et al. (2020) in southern Labrador and the Irminger Sea. Furthermore, similar concentrations of $^{129}$I and $^{236}$U between BBMW and LSW (Fig. 4)

suggest that substantial mixing also occurs between these water masses. With an average salinity 34.49 g/kg, BBMW can also act as a further freshwater source to LSW. These previously unreported freshwater contributions to LSW from Transition Water and BBMW can help shed light on potentially unresolved physical processes in ocean models that contribute to deep convection variability in the Labrador Sea.

Finally, NEADW is a water mass that originates from multiple sources, including ISOW, DSOW, and LSW (Yashayaev,

2007; García-Ibáñez et al., 2015). In the mixing model (yellow triangles in Fig. 6C), NEADW occupies a low-tracer area influenced by several of these sources. However, the elevated $^{236}$U relative to $^{129}$I is not explained by the endmembers established so far. Our results suggest that Transition Water and/or BBMW contribute up to 30% to NEADW, with the remaining from NAC waters. During autumn and winter, BBMW may enter the deep Labrador Sea as a weak overflow through Davis Strait (Huang et al., 2024; Curry et al., 2014) and join this deeper layer in the Labrador Sea, thus mixing with NEADW. Although the

simple two-endmember model used here cannot fully resolve every water mass contribution under low tracer concentrations and multiple sources, it provides a valuable first-order approximation. Future work could build on this foundation with a more comprehensive multiparameter analysis.



## 5 Conclusions

By applying $^{129}$I and $^{236}$U as tracers in Baffin Bay, Davis Strait, and the Labrador Sea, this study provides new constraints on the pathways and transformations of Arctic and Atlantic waters. Elevated $^{236}$U in Transition Water reveals that Arctic–Atlantic Water from the Canada Basin contributes significantly to its formation, demonstrating that the outflow of Atlantic-derived waters, most probably through Lancaster Sound, has been underestimated so far. At the same time, our results show that the water from the West Greenland Shelf Water feeds into the central Baffin Bay Arctic Water, while the cold Arctic Water largely originates from Nares Strait and mixes with West Greenland Shelf water. These processes highlight the role of Arctic outflows in the delivery of freshwater to Baffin Bay and further south. Beyond the Davis Strait, our results show that Transition Water provides a substantial contribution to the formation of Labrador Sea Water. This previously unrecognised pathway not only supplies freshwater but also leaves a distinct tracer signature, with direct implications for convection processes in the Labrador Sea and the composition of North East Atlantic Deep Water.

Together, these findings emphasise that Arctic outflows through the Canadian Arctic Archipelago are more important than previously assumed. They shape the transformation of Baffin Bay water masses and exert a significant influence on the ventilation and freshwater budget of the subpolar North Atlantic. Capturing these processes more accurately in ocean models will be essential for predicting future changes in deep water formation and the stability of the Atlantic Meridional Overturning Circulation.

*Data availability.* The original datasets for this study can be found at the Zenodo database (https://doi.org/10.5281/zenodo.16914587). The CTD data of Lancaster Sound and Nares Strait are provided by Amundsen Science Data Collection Amundsen Science Data Collection (2024).

*Author contributions.* LGTL, MC and NC contributed to the conception and design of the study. LGTL performed the data investigation, formal analysis, and wrote the original draft of the manuscript. The investigation and formal analysis were supported by NC and MC. JL contributed substantially to the interpretation of the results. HT, CV and NC performed AMS measurements. NC and MC supervised the study. NC and LGTL acquired funding. KAS, CL and MR supported sample collection during expeditions and provided hydrographic data. All authors contributed to the manuscript revision and approved the submitted version.

*Competing interests.* The authors declare that the research was without a conflict of interest.

*Acknowledgements.* The PI Núria Casacuberta has received funding from the European Research Council (ERC) under the European Union's Horizon 2020 research and innovation programme (TITANICA project, ERC2020-COG 101001451) and from the Swiss National Science



580   Foundation (PR00P2-193091-TRACEATLANTIC). In addition, Lisa Leist received funding from the Swiss Polar Institute (Polar Access Fund, PAF-2022-03). Jed Lenetsky received funding from the U.S. National Science Foundation awards 1902628 and 1902595. KAS is funded through the OFSI fund, Davis Strait Observing System, by Fisheries and Oceans Canada. The authors acknowledge the chief scientists, the captains and the crew of the R/V Neil Armstrong and the R/V Atlantis. The scientists involved in the sampling are deeply acknowledged. The authors also thank Emmy Hieronimus and Catherine Jeandel, who collected the samples at the Lancaster Sound and Nares Strait onbord

585   of CCGS Amundsen. Kayley Kündig and Simona Staub are thanked for their contributions to ETH-based laboratories. In addition, the authors acknowledge the entire ASOF community and their contribution through various discussions. Maps were created using cartopy and the GEBCO data product.





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

815





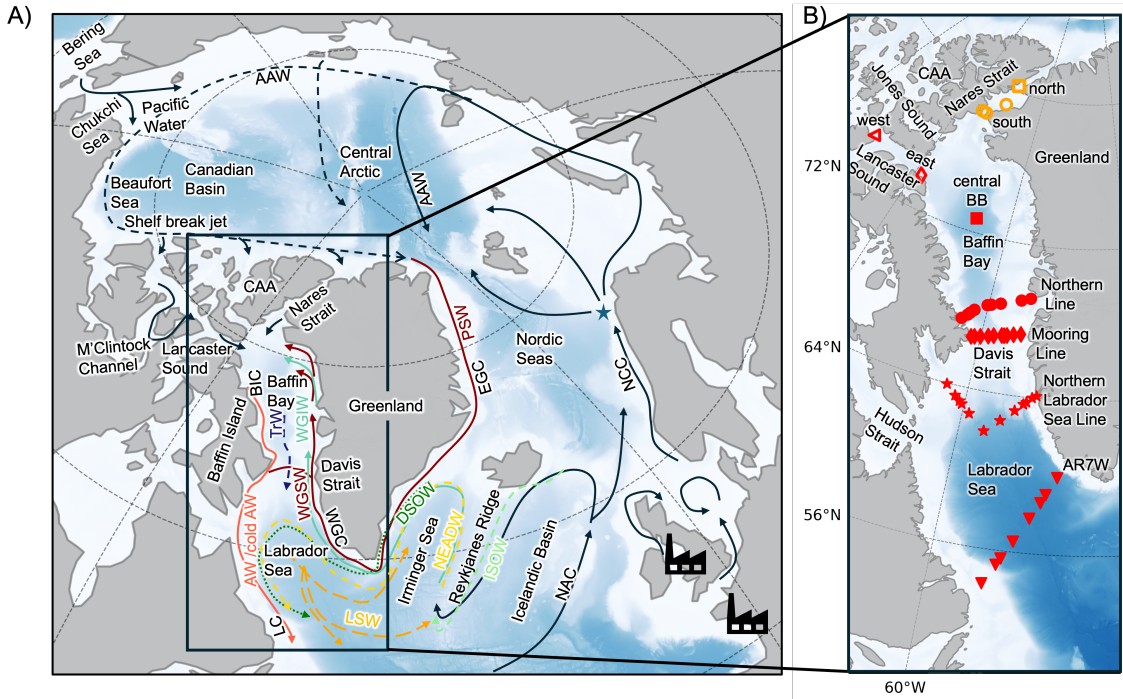

**Figure 1. A)** Map of the study area showing the main water mass circulation, adapted from Curry et al. (2014) and Dale et al. (2024). Black icons mark the locations of the nuclear fuel reprocessing plants at Sellafield (UK) and La Hague (France). **B)** Close-up view highlighting regional geographic details and the oceanographic transects sampled in this study, indicated by distinct red and orange symbols. All acronyms are defined in Appendix A1.



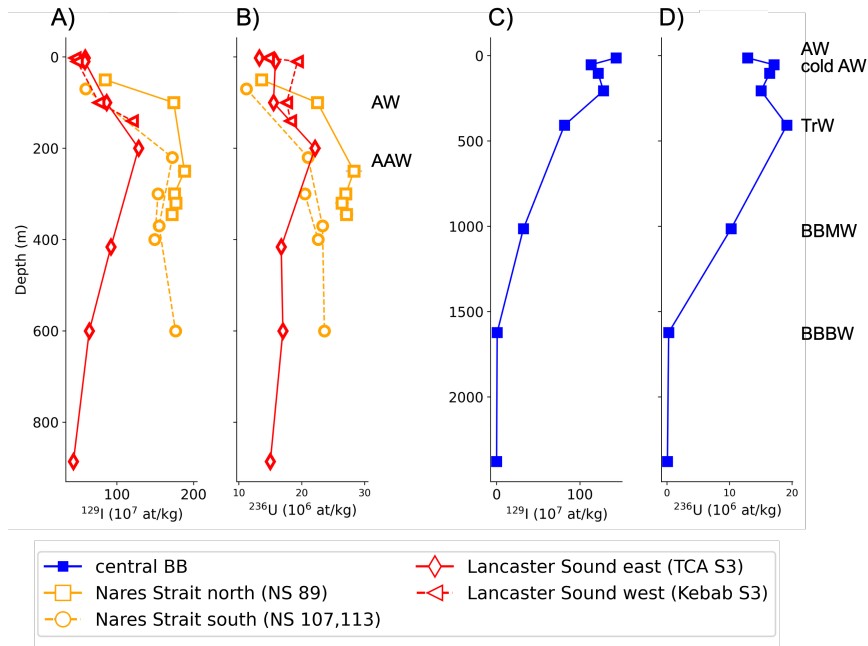

**Figure 2. A**) Concentrations of $^{129}$I in the Nares Strait (orange) and Lancaster Sound (red) profiles. **B**) Concentrations of $^{236}$U in Nares Strait and Lancaster Sound; **C**) $^{129}$I concentrations in the Central Baffin Bay station and **D**) Concentration of $^{236}$U in the Central Baffin Bay Station.



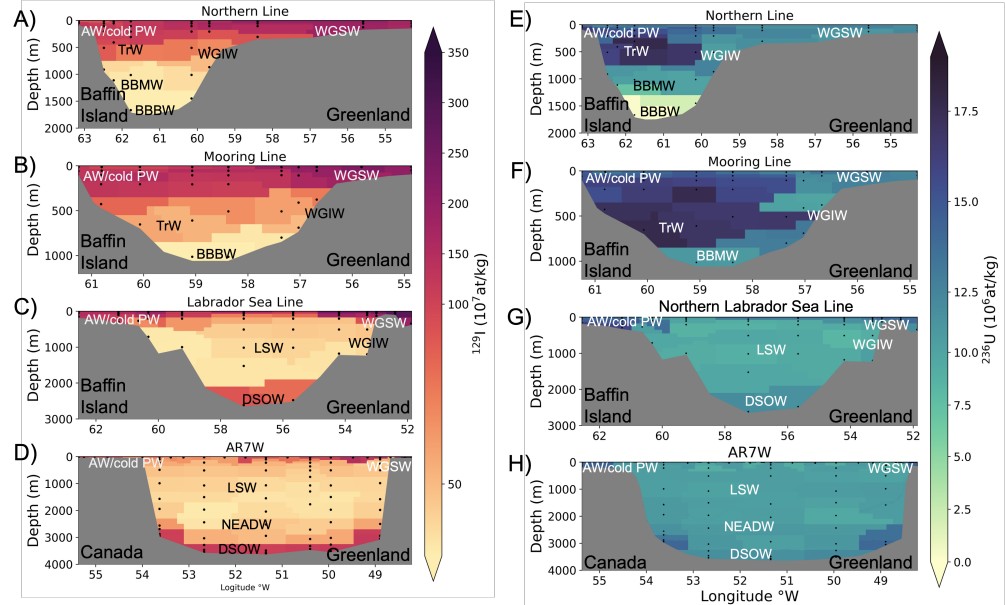

**Figure 3.** Zonal distribution of $^{129}$I (**A-D**) and $^{236}$U concentrations (**B-H**) along Northern Line (**A, E**), Mooring Line (**B, F**), Northern Larador Sea Line (**C, G**) and AR7W (**D, H**) respectively, in 2022. The water mass are based on Curry et al. (2014); Yashayaev (2007); Huang et al. (2024).



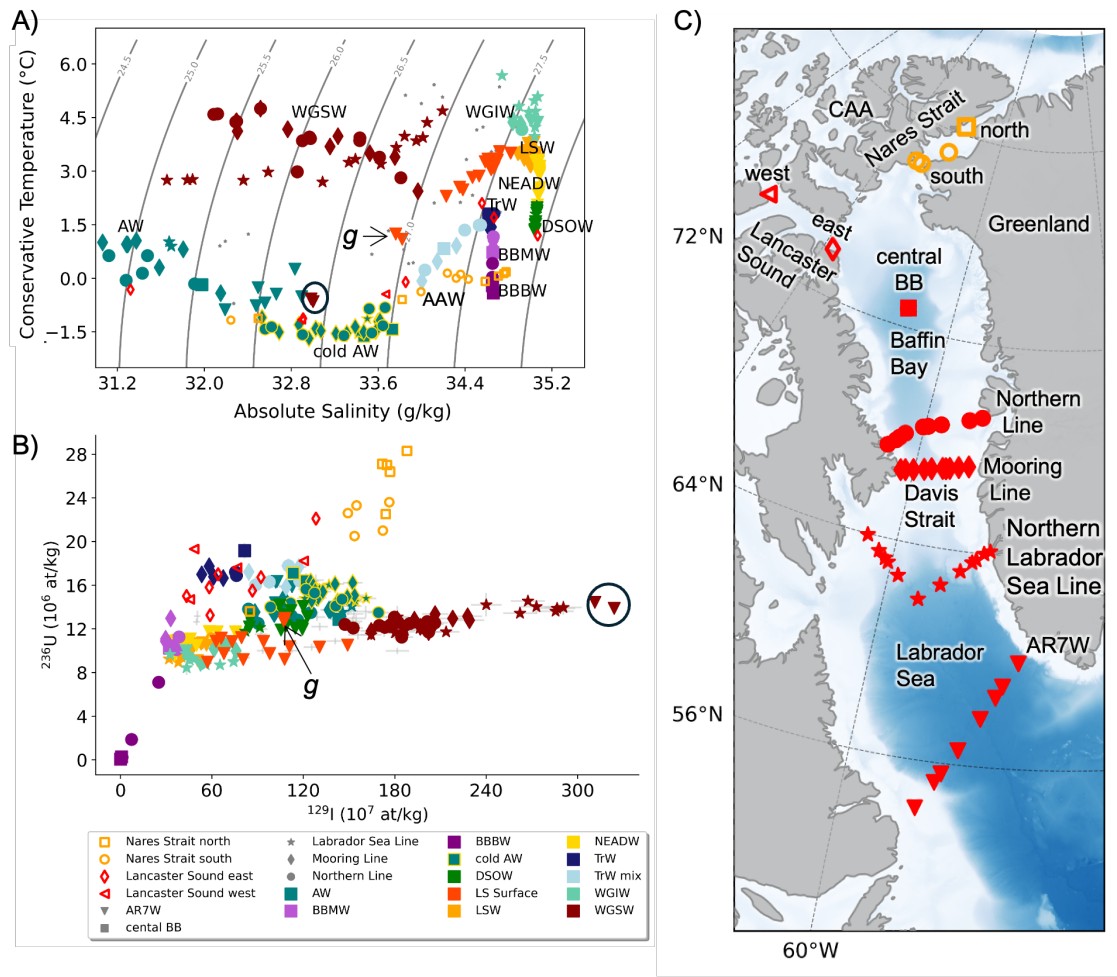

**Figure 4.** Overview of hydrographic and tracer data. **A)** Temperature-salinity (T-S) diagram, showing the distribution of water masses. **B)** $^{129}$I versus $^{236}$U plot for all samples. Water masses are equally colour coded both in A and B. **C)** Map of the study area with symbols marking the sampling locations; the symbols correspond to those used in panels A and B to indicate sampling provenance. Black circles highlight two surface samples discussed in Section 4.5, while samples highlighted by "g" are discussed in Section 4.5.



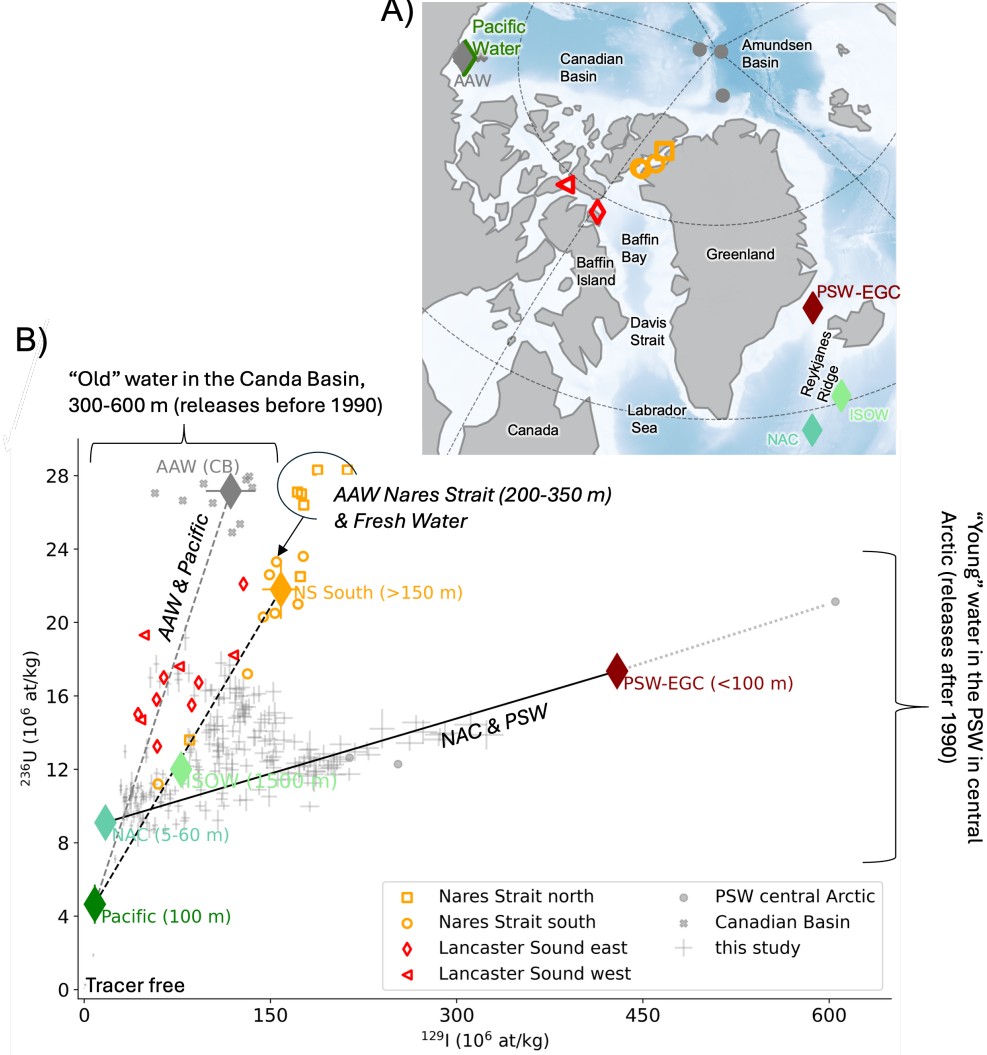

**Figure 5. A)** location of each endmember and the samples in the Arctic Ocean, used to derive the endmembers. **B)** $^{129}$I - $^{236}$U tracer space with endmembers indicated by diamonds and samples reported in this study shown with light grey crosses. Published data from the Arctic Ocean is indicated by crosses and circles. Data from Lancaster Sound (red symbols) and Nares Strait (orange symbols) have not been published before. AAW (CB): Arctic Atlantic Water at Canada Basin (Payne et al., 2024), NAC: North Atlantic current (Castrillejo et al., 2018), PSW-EGC: Polar Surface Water at the East Greenland Current (Dale et al., 2024), Pacific Water (Payne et al., 2024), PSW central Arctic: Polar Surface Water in the central Arctic (Wefing et al., 2025). Tracer free: very old waters without anthropogenic signature, glacier and sea ice melt, river runoff, and precipitation.





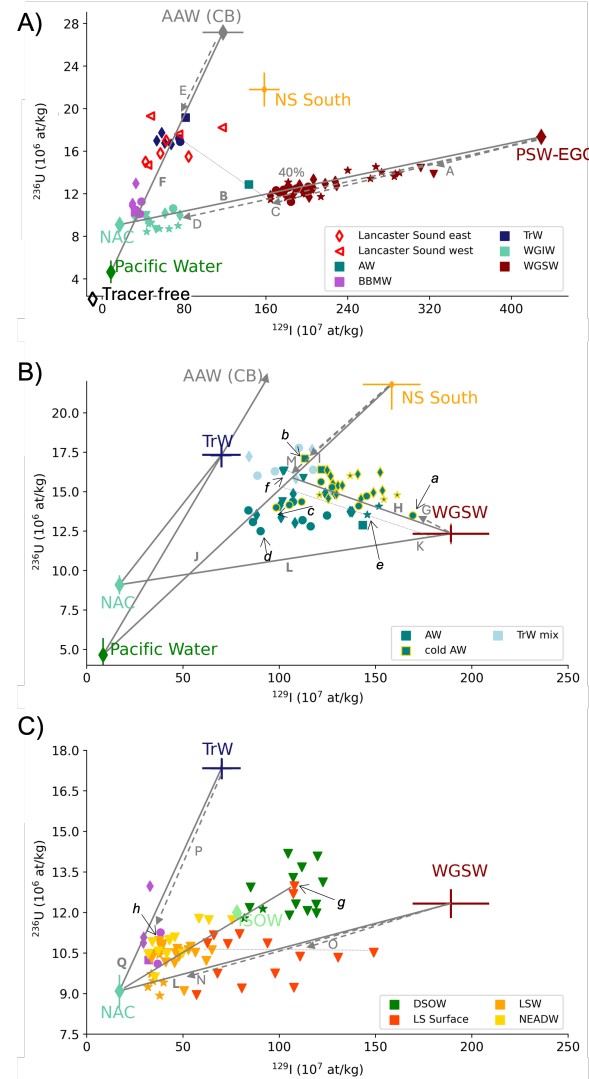

**Figure 6.** Binary mixing of $^{129}$I versus $^{236}$U used to infer water mass origins and mixing. Symbol shape indicates sampling locations: central Baffin Bay (square), Northern Line (dots), Mooring Line (diamonds), Northern Labrador Sea Line (stars), and AR7W (triangles). Colored diamonds mark endmembers, with crosses showing endmembers defined in this study, including their maximum spread. The solid line represents mixing between endmembers; dashed lines show each endmember's contribution to a sample, and dotted lines interpolate samples to the mixing line. Capital letters label distances for mixing fraction calculations (see Section 4), while lowercase letters (a–h) highlight samples discussed further. Panels show binary mixing models for: **A)** Transition Water (TrW), West Greenland Shelf Water (WGSW), West Greenland Irminger Water (WGIW), and Baffin Bay Mode Water (BBMW); **B)** cold Arctic Water (cold AW), Arctic Water (AW), and Transition Water mix; **C)** Labrador Sea Water (LSW), Labrador Sea Surface water, and North East Atlantic Deep Water (NEADW). References, sample counts, and sampling years for endmembers are listed in Appendix Table A2. Acronyms are explained in Appendix A Table A1







**Figure 7.** T—S diagram with colored $^{236}$U concentrations **A)** along the Mooring and Northern Line, and in central Baffin Bay. **B)** The single stations at Nares Strait and Lancaster Sound.




**Table A1.** Appendix A: Acronyms, Water Masses, and Currents: The water masses referenced in this study are based on classifications from Curry et al. (2014), Huang et al. (2024), García-Ibáñez et al. (2015), Yashayaev (2007) and Bourke et al. (1989). In this work, hydrographic properties are reported as follows: Absolute Salinity ($S_A$, g/kg) and Conservative Temperature (CT, °C) are given for all data newly analysed in this study, following TEOS-10 recommendations. For water masses described in the literature whose properties were not re-evaluated here, values are reported as practical salinity (PSU) and potential temperature ($T_{pot}$, °C), following the sources. This approach ensures consistency with both current standards (TEOS-10) and historical literature for all water mass and current definitions presented.

| Acronym | Water mass/ current | Salinity | Temperature | Reference |
|---|---|---|---|---|
| AAW (CB) | Arctic Atlantic Water Canada Basin | >34.7 | $T_{pot}$>0 | Payne et al. (2024) |
| AAW | Arctic Atlantic Water | $S_A$>33 | CT>-1 | this study |
| AW | Arctic Water | $S_A$>32.9 | 1.1>CT>-0.9 | this study |
| BBBW | Baffin Bay Bottom Water | $S_A$<34.6 | CT<0 | this study |
| BBMW | Baffin Bay Mode Water | $S_A$>34.6 | 0.7<CT<1.2 | this study |
| cold AW | cold Arctic Water | 32.5<$S_A$<33.8 | CT<0.8 | this study |
| DSOW | Denmark Strait Overflow Water | $S_A$<35 ±0.1 | CT<1.3 | this study |
| ISOW | Iceland–Scotland Overflow Water | 34.9-35 PSU | $T_{pot}$:2.0-3.5 | Dale et al. (2024) |
| LSW | Labrador Sea Water | $S_A$<34.9 | 3.8>CT>3.0 | this study |
| NEADW | North East Atlantic Deep Water | 35.06±0.2 | 2.4<CT<3.3 | this study |
| Pacific Water | Pacific Water | <32.5 PSU | $T_{pot}$-0.4±0.1 | Payne et al. (2024) |
| PSW central Arctic | Polar Surface Water | density: $\sigma_\theta$ <27.70 | | Wefing et al. (2025) |
| PSW-EGC | Polar Surface Water at Denmark Strait | <34.3 PSU | $T_{pot}$<0 | Dale et al. (2024) |
| SAIW | Subarctic Intermedaite Water | 34.9 PSU | $T_{pot}$:4-7 | Castrillejo et al. (2018) |
| TrW | Transition Water | $S_A$: 34.64±0.2 | CT<0.8 | this study |
| TrW mix | Transition Water mix | $S_A$:33.8-34.7 | 1.4<CT<1.8 | this study |
| WGIW | West Greenland Irminger Water | $S_A$ >34.4 | CT>3.5 | this study |
| WGSW | West Greenland Shelf Water | $S_A$<34.2 | CT<5 | this study |
| BIC | Baffin Island Current | | | |
| CAA | Canadian Arctic Archipelago | | | |
| EGC | East Greenland Current | | | |
| LC | Labrador Current | | | |
| NAC | North Atlantic Current | | | |
| NCC | Norwegian Coastel Current | | | |
| WGC | West Greenland Current | | | |

## A1 Appendix A, Acronyms





**Table A2.** Endmember Data with Geographic, Isotopic, and Hydrographic Properties

| Endmember | Latitude °N | Longitude °E | $^{129}$I $\times10^7$at/kg | $^{236}$U $\times10^6$at/kg | $S_A$ | CT | # Samples | Ref |
|---|---|---|---|---|---|---|---|---|
| AAW CB | 70.6 - 77.7 | -140.0 - -146.8 | 118 ± 20 | 27.2 ± 0.7 | 34.96 ± 0.06 | 0.5 ± 0.2 | 13 | a |
| Pacific | 70.6 - 77.7 | -140.0 - -146.8 | 8.5 ± 2.4 | 4.6 ± 1.1 | 32.13 ± 0.25 | -0.7 ± 0.4 | 8 | a |
| PSW-EGC | 67.5 | -25.8 | 429 ± 5 | 17.3 ± 0.3 | 33 ± 1 | 1.2 ± 3.8 | 2 | b |
| NAC | 58.5 | -31.2 | 16.93 ± 0.4 | 9.1 ± 0.6 | 35.23 ± 0.01 | 8.8 ± 0.7 | 3 | c |
| ISOW | 55.3 | -26.4 | 78 ± 2 | 12 ± 0.1 | 35.14 ± 0.01 | 2.7 ± 0.1 | 3 | b |
| NS South | 78.3 | -73.3 - - 74.7 | 158 ± 15 | 21.8 ± 1.6 | 33.9 ± 0.4 | -0.1 ± 0.4 | 6 | d |
| WGSW | 63.7 - 69.2 | -53.0 - -58.4 | 189 ± 20 | 12.3 ± 05 | 33.2± 0.6 | 3.8 ± 0.6 | 30 | d |
| TrW | 66.7 - 72.7 | -57.3 - -65.9 | 70 ± 10 | 17 ± 0.1 | 34.6 ± 0.2 | 1.6 ± 0.2 | 7 | d |

*PSW-EGC corresponds to PSW at Denmark Strait, a:Payne et al. (2024), b:Dale et al. (2024), c:Castrillejo et al. (2018), d: this study.*





## A2 Appendix B, Additional Figures

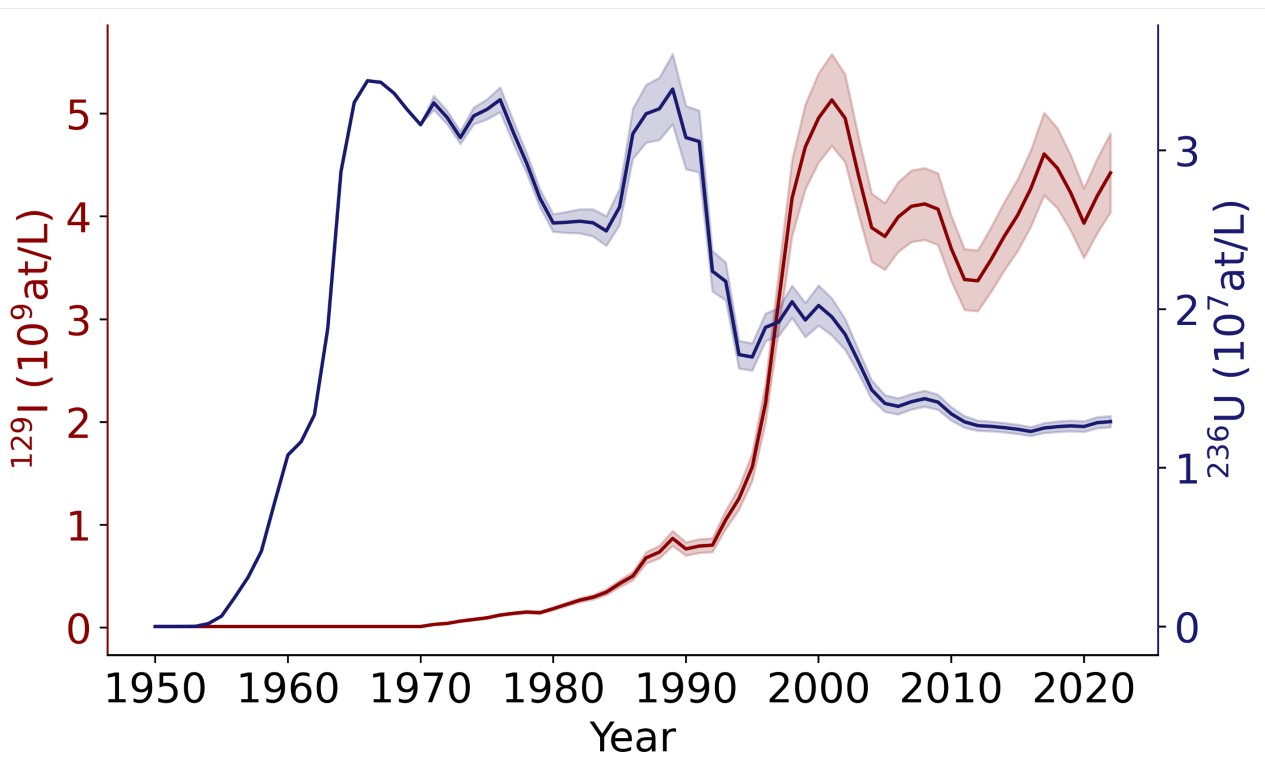

**Figure A1.** Input function of $^{129}$I (red) and $^{236}$U (blue) defined at 70°N showing the combined imput from nuclear fuel reprocessing plants in Sellafield (UK) and La Hague (F), and the global fallout from atmospheric nuclear weapon tests.





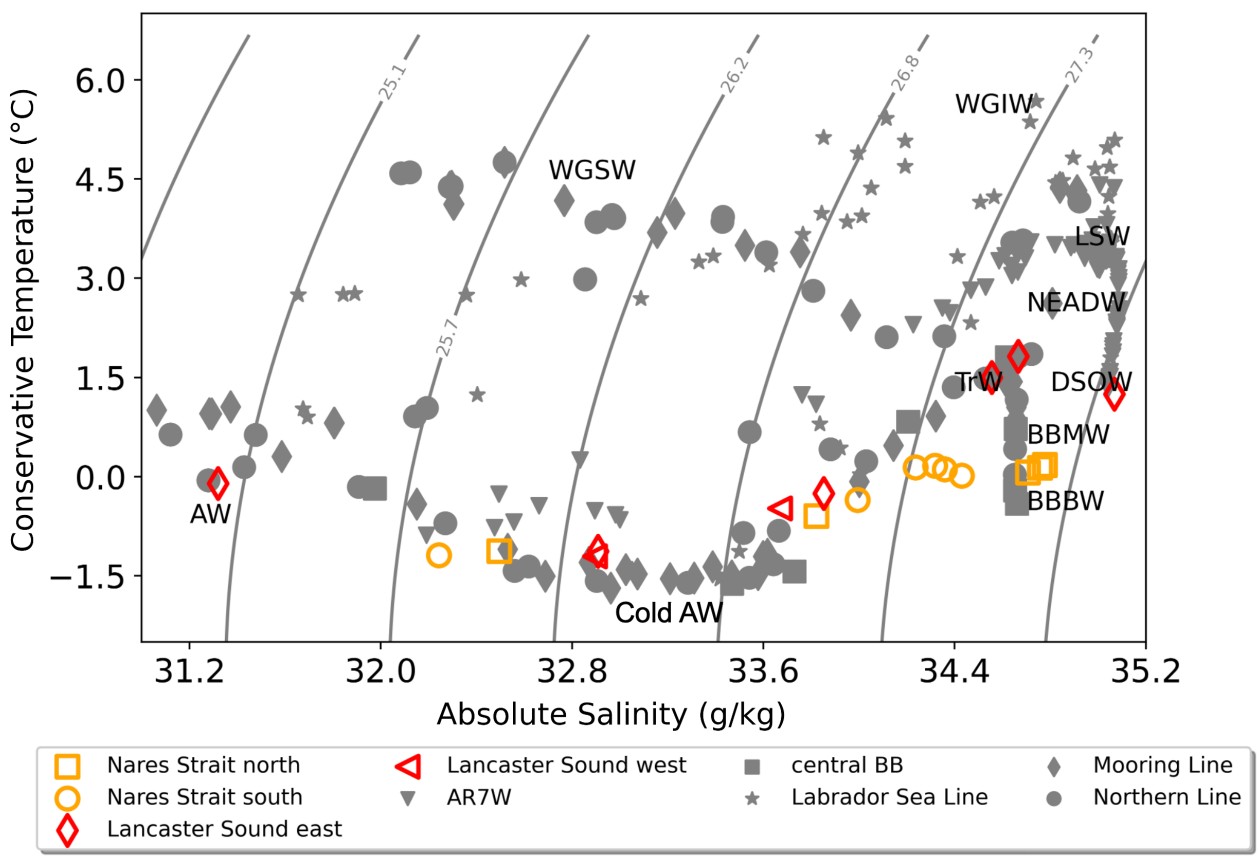

**Figure A2.** T—S diagram, of all samples reported in this study. Grey symbols represent the samples covering the four lines and central Baffin Bay, sampled in 2022. The red symbols represent samples from Lancaster Sound, while the orange symbols represent the samples from Nares Strait, both sampled in 2024.



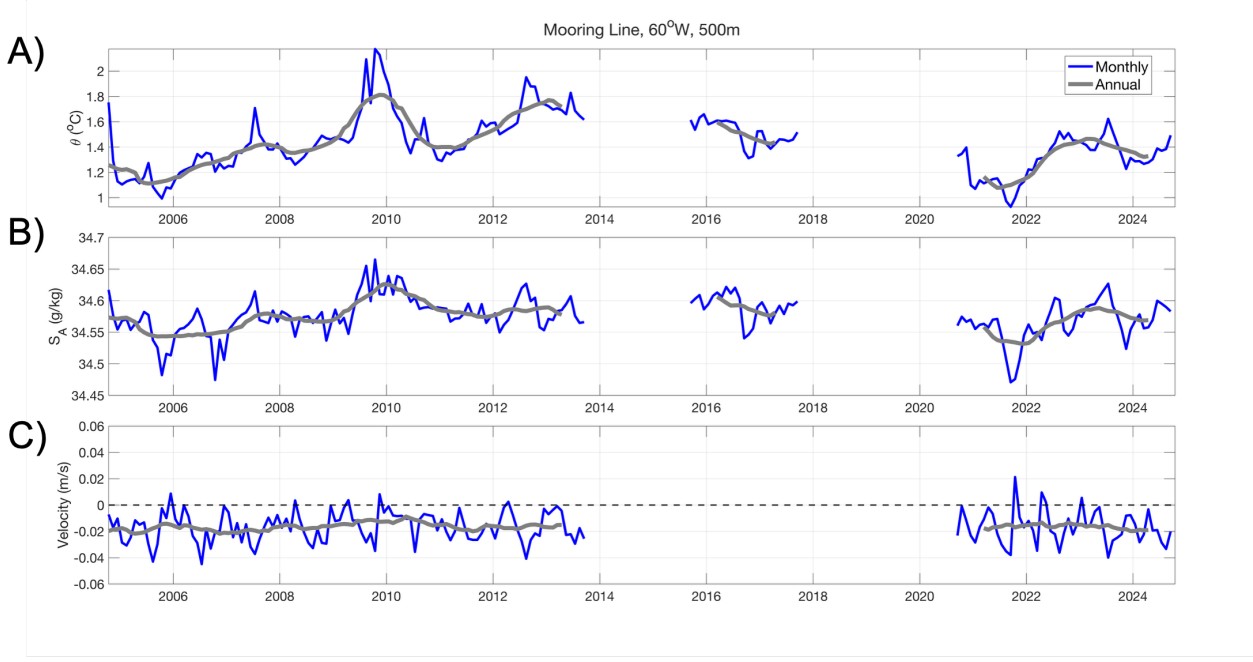

**Figure A3.** Time series of **A**) Conservative Temperature ($\theta$), **B**) absolute salinity ($S_A$), and **C**) across-strait velocity, at $60°$W and 500 meters at the Mooring Line. The blue line shows the 30-day running mean and the grey line shows the 12-month running mean.





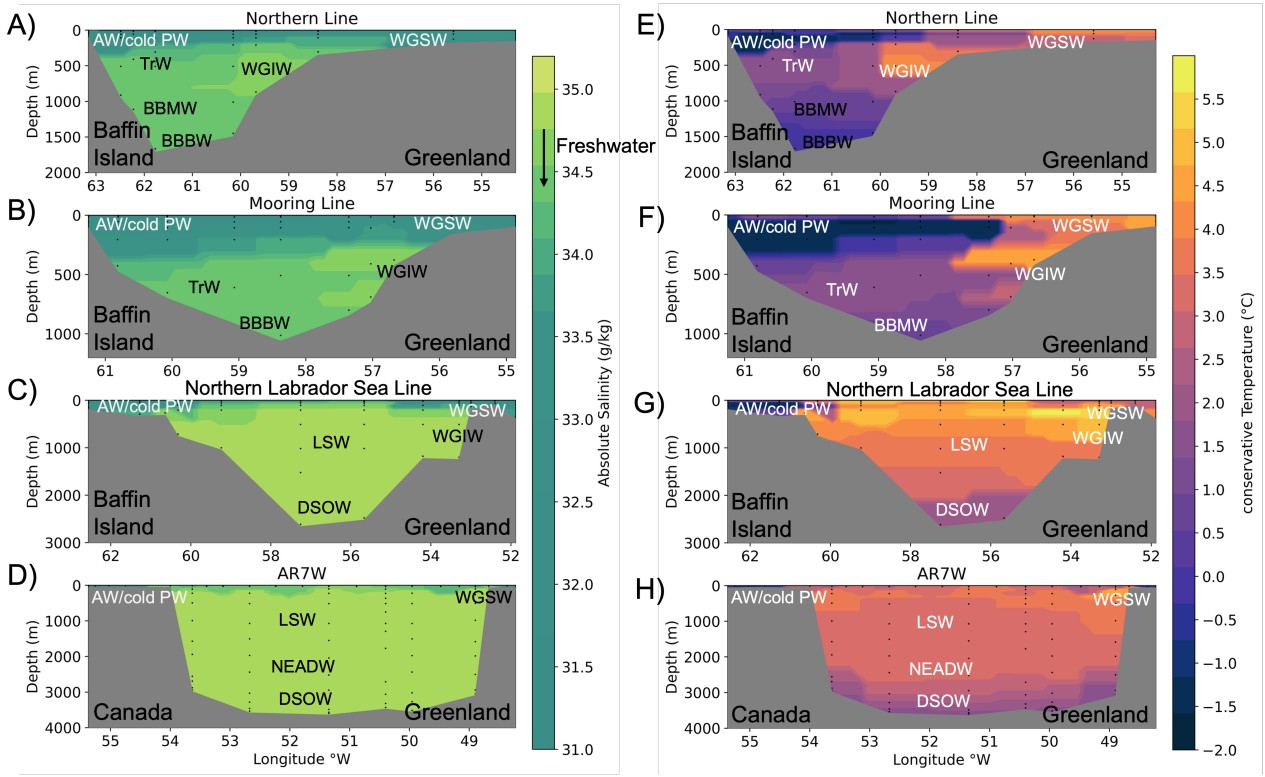

**Figure A4.** Section Plots of Conservative Temperature (**A-D**) and Absolute Salinity (**E-F**), along the Mooring Line (A, D), Northern Line (B-F), Northern Labrador Sea Line (C, G) and AR7W Line (D, H).