# Peer review of "Radionuclide tracers reveal new Arctic pathways shaping water mass mixing and formation in Baffin Bay and Labrador Sea"

_EGUsphere, 2025_

## Referee Comment (RC1)

Review of article: Radionuclide tracers reveal new Arctic pathways shaping water mass mixing and formation in Baffin Bay and **the** Labrador Sea

**General Comments - paragraph evaluating overall quality of the article**

This paper utilizes measurements of the artificial radionuclides 129I and 236U from seawater in Baffin Bay and the Labrador Sea, in order to trace the origins and interactions of various water masses present in the region. The use of these artificial radionuclides as water mass tracers can provide new insights compared to previous tracer studies in the region that have employed nutrient concentrations (e.g. the nitrate to phosphate ratio), stable oxygen isotopes, as well as temperature and salinity measurements. This study provides a substantial contribution to furthering knowledge of water mass formation and interactions between Arctic-outflow waters from the Canadian Arctic Archipelago and the subpolar North Atlantic.

One general comment that I have is that the calculation of mixing fractions is not explained very clearly. I think that simply stating that a "mixing fraction = length A/B" without clearly defining what A and B are (points? Line segments?), is not thorough enough for a scientific paper. I would be happy to see clearer language used around the calculation of mixing fractions (I provide an example sentence from Leist et al. (2024) that I think is more clear, in my other specific comments). As another option, providing some example equations within the methods section could also help. This clarity issue comes up again throughout the discussion section, and I have made specific comments below about where I think the text lacks clarity in regards to this.

A minor general comment that I have, throughout the manuscript, is that when referring to smaller passageways or water bodies in the Canadian Arctic, such as Davis Strait, Lancaster Sound, Nares Strait, and Baffin Bay, you do not need to include "the" before their names in sentences. However for Seas or Oceans, "the" should be included before their names (e.g., the Labrador Sea, the North Atlantic Ocean). The authors are currently inconsistent about this throughout the text, please check for consistency.

**Specific comments:**

Paragraph 2 of the Introduction (lines 32 to 38): I think it should be clarified here that the West Greenland boundary current system consists of two components: (1) the West Greenland Coastal Current (WGCC), which transports fresh and relatively warm PSW from the Arctic Ocean, and represents the continuation of Arctic outflow waters from the EGC, and (2) the West Greenland Current (WGC) which transports warm and salty WGIW at depth, and is a shelfbreak jet (see introduction of Huang et al. 2024 for details on this distinction). It is the WGC that bifurcates into two branches in the northern Labrador Sea, with the majority of the current being diverted westwards across the northern Labrador Sea, and only a limited amount continuing north into Baffin Bay. I suggest adding these details into your introduction.

Line 85-86: I would also suggest citing Bamber et al. (2018) "Land ice freshwater budget of the Arctic and North Atlantic Oceans: 1. Data, methods and results" JGR: Oceans

Line 165-167: I am a bit confused by this sentence – are both stations NS89 and NS79 represented by the orange square in northern Nares Strait in Fig. 1B? And the orange circles represent southern Nares Strait stations NS102 to NS114 and NS107? This could be phrased more clearly.

Last paragraph of section 2.2 (lines 190 - 203): Many of the water mass definitions in this paragraph do not match the defined ranges of conservative temperature (CT) and absolute salinity (SA) given in Table A1. Please make sure they match for consistency, or where they may differ provide some clarification as to why. For example, in Table A1 West Greenland Irminger water (WGIW) is defined as CT > 3.5 and SA < 34.2 but in this paragraph it is defined as CT > 4 and SA > 34.7.

Line 211: I understand the concept of how mixing fractions are calculated along mixing lines between end-member values, but I don't find it to be very clear from this sentence. Especially the "(fraction = length A/B) with A and B being end-member values". Since this is not a real equation, just the concept of it that you are trying to convey. I looked at Leist et al. (2024) and I preferred this sentence from that paper, I find this more clear: "The water mass fraction is calculated by dividing the distance between the sample and one endmember by the total distance between the two endmembers". I think this wording gives the reader a better idea about how to re-produce the method.

Line 212: In this sentence do you mean that end-members were calculated as the mean of sample values in a certain CT and SA range? This is stated a bit more clearly on lines 215-216, but I think it should be stated earlier in this first paragraph of section 2.3. And how exactly were the estimated fractions rounded? I think this sentence could be phrased more clearly.

Line 292-294: The part of this sentence that states "...with a clear influence from low-tracer Pacific water" didn't make sense to me, because a Pacific water end-member is not shown in tracer space on Figure 4. It made sense once I saw the Pacific end-member in Figure 5 tough.

Section 4.1.1: As I was reading the first paragraph of this section, describing how the various end-member values were defined, I was wondering about the depth range used to define each end-member. For example, the NAC end-member and ISOW are both defined in the region east of the Reykjanes Ridge, but they have very different depth ranges (and T/S properties; Table A1). I see that the depth ranges for each end-member are listed on Figure 5B – perhaps just a mention in the text that the depth ranges relevant to each end-member can be found in Figure 5B would be nice.

Line 350: This sentence mentions a diamond with a black outline in Fig. 5B – but the symbol is missing from the figure.

Line 374: Here it is stated "fraction PSW-EGC = A/B, Fig. 6A, see section 2.3 for fraction calculation". I can see that the capital letters for A and B are on Figure 6A, but it looks like A, C, and D represent the length of dashed arrows? And I assume that B is supposed to represent the entire length of the line between the NAC and PSW-EGC end-members? I would appreciate if the definitions of A, B, and other letters could be clearly defined in the text for the reader, so that there is no confusion about the calculation. I think this also needs to be clarified earlier in the methods section 2.3.

Lines 377-378: This sentence states "...consistent with estimates from Huang et al. (2024) for southern Baffin Bay (160 m out of 600 m)...". I'd like some clarify on what these depths indicate – I

believe you are referring to the thickness of the WGSW layer in the total water column, as was presented in Huang et al.? Please clarify what these depths are indicating.

Line 380: This is again in reference to my earlier comment that I would like it to be clearly stated what C and B represent.

Line 387: In this sentence the fraction of PSW-EGC = **1-**D/B. Why is this formula for the fraction of PSW-EGC different from those at line 380 and 374?

Line 389: Similar to my earlier comment about depths stated from Huang et al. (2024) – what do these depths represent?

Section 4.3 title: Should this title say Baffin Bay Mode Water, instead of bottom water? Since BBMW is discussed in the final paragraph of this section. I would be interested to read more about the interpretation of the very low tracer concentrations of BBBW though.

Line 409: Again, could E and F be defined more clearly in the text? And would the fraction of AAW (CB) be equal to 1-E/F? I have the same comment for Lines 463, 467, 486, 488, 510, 511 and 532 in relation to Fig. 6B and 6C. Line 486 does provide a bit more clarification that "L" represents a line, maybe K should be a dashed arrow similar to G?

Line 481: Doesn't cold Arctic water show the strongest WGSW influence? As evidenced by point "a" in the middle of the Northern Line (65% WGSW), as stated on line 462. Whereas the greatest WGSW contribution to Arctic water at point "e" is 30%.

Line 509 and 525: This sentence states "Labrador Sea Water (red symbols Fig. 6C)..." I am unsure if you are referring to the points labelled as LSW or LS Surface in Fig/ 6C? To me, the LSW triangles appear an orange color, and LS Surface appears a dark orange.

Line 520-522: I believe another possible mechanism that might lead these two surface samples (indicated by "g" in Fig. 6C) to have relatively higher 236U (in the range of DSOW) is a contribution of cold AW. Rysgaard et al. (2020) showed evidence for a southward current along the southwest Greenland coast, which transports Baffin Bay Polar Water (referred to as cold AW in this manuscript) as far south as 64 degN latitude. I see that you already consider/discuss that Arctic water and cold Arctic water from the Labrador Current would only likely play a very small role (lines 523-524), but what if there is another southern transport on the east side of Davis Strait?

**Technical corrections:**

For the title of this article, it should end with "the Labrador Sea".

Line 37-39: I found this sentence difficult to read as currently worded, I suggest changing it to: "In the northern Labrador Sea it bifurcates into two branches, with one continuing north into Baffin Bay, and another (larger) branch following the bathymetry of the Labrador Sea, turning westwards towards the Labrador Shelf."

Line 42: The BIC current in Figure 1A appears light red in color to me, with the LSW appearing orange, and the NEADW yellow. The WGSW current is dark red.

Line 43: "transports fresh water of Arctic-origin" – Arctic should always be capitalized.

Line 48: Make it more clear that the depths listed in brackets are sill depths for each Arctic-outflow passageway

Line 50: "surface of Baffin Bay and along the BIC"

Line 60: should be "bathymetrically **steared**" not stirred.

Line 64: I think this line should read "...approximately 6-8% of LSW" to be more clear.

Line 116-117: I suggest changing some wording in this sentence to "...from the global fallout from nuclear weapons testing in the 1960s, and liquid releases from nuclear reprocessing plants in Sellafield (UK) and La Hague (France)..."

Line 184: Spelling should be "Zenodo" not Zenode.

Line 197: I believe the word "overlying" should be used here instead of "overflowing" to indicate that BBMW sits on top of BBBW.

Line 232: A range is given for 129I, but 236U is stated separately for north and south. Could also present 236U values as a range, since the reader can see which location has the lower/higher value in Figure 2.

Line 281: Should have a bracket before "Fig. 4A and B indicated by black circles)."

Line 302: The Labrador Sea surface water (LS surface) symbols in Figure 4 appear as a dark orange to me, while the LSW appears as a lighter orange.

Line 305: Should reference Fig 6C here I believe.

Line 310: Sentence should end with "concentrations", plural for both tracers.

Line 330: Spelling of "reprocessing".

Line 366: This sentence should say "...with cooling and salinification in winter and **freshening** in summer".

Line 390: I believe this sentence should say "...due to the limitation of the mixing **of** two endmembers in the tracer analysis."

Line 454: This sentence should say "For WGSW, the mean and standard deviation were derived **from...**"

Line 455: This sentence should say "Cold Arctic Water **generally has** higher tracer concentration**s**..."

Line 465: Spelling of polynya.

Line 497: I suggest rephrasing the statement to be "Therefore, AAW originating from either Lancaster Sound or Nares Strait needs to be considered."

Line 505: Sentence should read "In a previous study by Leist et al. (2024) using  $^{129}$ I and  $^{236}$ U **in** the Labrador Sea..."

Line 535: This sentence refers to the 236U maximum on the mooring line – perhaps reference Fig. 3 to support this statement, and the "x" symbol is missing in the given 236U value here (scientific notation).

Line 557: The sentence that starts on this line, I would rephrase as "At the same time, our results show that West Greenland Shelf Waters feed into Arctic waters in central Baffin Bay, while cold Arctic Waters largely originate from Nares Strait and mixes with West Greenland shelf water."

Line 560: Sentence starting on this line, I would re-phrase as "South of Davis Strait, our results show that Transition Water...". I would also consider mentioning in this sentence that TrW might also provide a contribution to NEADW.

**Technical corrections on Figures:**

Figure 1: In this figure caption I would state "Appendix **Table** A1", similar to the caption for Figure 6. Also, perhaps clearly state that ocean current names are stated in black text, while specific water mass names are in colors.

Figure 3: In panel (B) BBBW is labeled, but then in panel (F) BBMW is labeled in the same place. I am assuming this label should be the same between the two panels? Also the caption for Figure 3 should read "236U concentrations (**E**-H)" and "The water mass**es** are based on Curry et al..."

Figure 5: Check spelling of Canada Basin in the label on the top left of Fig. 5B.

Figure A1: Please indicate where the data for this plot was obtained – would also be helpful to reference the data source in the text somewhere between lines 115 – 120. Also, spelling of "input" in the figure caption is incorrect.

Figure A3: The lower-case theta symbol on the y-axis of panel A usually represents potential temperature, and conservative temperature is represented by a capital theta symbol. Should the figure caption state potential temperature instead of conservative temperature? Another suggestion for this figure is to add shaded ranges of the CT/SA bounds of Transition water, to clearly highlight how long these measurements fall within the definition of that water mass.

Figure A4: The figure caption should be adjusted to state that "Section plots of Absolute Salinity (A-D) and Conservative Temperature (E-H), along the Northern Line (A,E), Mooring Line (B-F), Northern Labrador Sea Line (C,G), and AR7W Line (D,H). Many of the panels were referenced incorrectly in the figure caption, please check.